**EMBO** *reports*

# *N*-acetylaspartate release by glutaminolytic ovarian cancer cells sustains protumoral macrophages

Alessio Menga[1,2,3] (iD), Maria Favia[2,4,†] (iD), Iolanda Spera[2,†], Maria C Vegliante[5,†], Rosanna Gissi[2,†], Anna De Grassi[2], Luna Laera[2] (iD), Annalisa Campanella[2], Andrea Gerbino[2], Giovanna Carrà[3,6], Marcella Canton[4,7] (iD), Vera Loizzi[8], Ciro L Pierri[2], Gennaro Cormio[8,9], Massimiliano Mazzone[1,3,10,*] (iD) & Alessandra Castegna[2,7,**] (iD)

## Abstract

Glutaminolysis is known to correlate with ovarian cancer aggressiveness and invasion. However, how this affects the tumor microenvironment is elusive. Here, we show that ovarian cancer cells become addicted to extracellular glutamine when silenced for glutamine synthetase (GS), similar to naturally occurring GS-low, glutaminolysis-high ovarian cancer cells. Glutamine addiction elicits a crosstalk mechanism whereby cancer cells release *N*-acetylaspartate (NAA) which, through the inhibition of the NMDA receptor, and synergistically with IL-10, enforces GS expression in macrophages. In turn, GS-high macrophages acquire M2-like, tumorigenic features. Supporting this *in vitro* model, *in silico* data and the analysis of ascitic fluid isolated from ovarian cancer patients prove that an M2-like macrophage phenotype, IL-10 release, and NAA levels positively correlate with disease stage. Our study uncovers the unprecedented role of glutamine metabolism in modulating macrophage polarization in highly invasive ovarian cancer and highlights the anti-inflammatory, protumoral function of NAA.

**Keywords** IL-10; metabolism; *N*-acetylaspartate; ovarian cancer; TAMs
**Subject Categories** Cancer; Metabolism

## Introduction

The mechanisms underlying tumor growth, cancer cell invasion, and metastasis (Kolonin, 2012; Arvizo *et al*, 2013; Dragosavac *et al*, 2013) are often associated with cancer cell adaptation toward specific nutrient preferences for growth, invasion, and energy metabolism (Yang *et al*, 2014). These metabolic preferences are often dictated by oncogenic alterations associated with tumorigenesis (Dang *et al*, 2009; Gao *et al*, 2009; Weinberg *et al*, 2010; Gaglio *et al*, 2011) but also to the acquisition of a malignant state, which is dependent on cancer stage (Nomura *et al*, 2010; Benjamin *et al*, 2012; Caneba *et al*, 2012; Agus *et al*, 2013). Among the different nutrients, glutamine (Gln) is a key metabolite for cancer cell growth, as this amino acid becomes essential for "addicted" cancer cells (Weinberg *et al*, 2010; Wise & Thompson, 2010; Le *et al*, 2012; Castegna & Menga, 2018). This dependence relies on the many crucial roles Gln plays in the cell. Besides sustaining anaplerotically the TCA cycle, Gln represents a carbon and/or nitrogen source for other amino acids, fatty acids, and nucleotides (DeBerardinis & Cheng, 2010; Wise & Thompson, 2010; Rajagopalan & DeBerardinis, 2011; Daye & Wellen, 2012; Metallo *et al*, 2012). It is also the main source of *N*-acetyl glucosamine, necessary for protein glycosylation (Spiro, 2002), and of glutathione (Sappington *et al*, 2016), which endogenously detoxifies the cell from harmful radical species (Lauderback *et al*, 2003; Forman *et al*, 2009). Finally, Gln is emerging as an important modulator of many signaling pathways, such as mTOR, a regulator of autophagy (Van Der Vos *et al*, 2012) and protein synthesis (Nicklin *et al*, 2009). A positive correlation between Gln dependence of ovarian cancer cells and tumor aggressiveness has been extensively elucidated (Yang *et al*, 2014).

1  Department of Molecular Biotechnologies and Health Sciences, University of Turin, Turin, Italy
2  Department of Biosciences, Biotechnologies and Biopharmaceutics, University of Bari, Bari, Italy
3  Molecular Biotechnology Center, Turin, Italy
4  Department of Biomedical Sciences, University of Padova, Padova, Italy
5  Haematology and Cell Therapy Unit, IRCCS-Istituto Tumori 'Giovanni Paolo II', Bari, Italy
6  Department of Clinical and Biological Sciences, University of Turin, Orbassano, Italy
7  Fondazione Istituto di Ricerca Pediatrica Città della Speranza - IRP, Padova, Italy
8  Policlinico University of Bari "Aldo Moro", Bari, Italy
9  Gynecologic Oncology Unit, IRCCS, Istituto Tumori Giovanni Paolo II, Bari, Italy
10 Laboratory of Tumor Inflammation and Angiogenesis, Center for Cancer Biology, Department of Oncology, KU Leuven, Leuven, Belgium
   *Corresponding author. Tel: +32 16 37 32 13; E-mail: massimiliano.mazzone@kuleuven.vib.be
   **Corresponding author. Tel: +39 080 5442322; E-mail: alessandra.castegna@uniba.it
   †These authors contributed equally to this work

Gln metabolism is fundamental also for tumor-associated macrophage (TAM) function, as macrophage-specific targeting of glutamine synthetase (GS) in tumor-bearing mice skews TAMs and MAMs (namely metastasis-associated macrophages) toward an "M1-like" state, promoting tumor vessel pruning, vascular normalization, accumulation of cytotoxic T cells, and metastasis inhibition (Palmieri *et al*, 2017; Menga *et al*, 2020). Furthermore, GS is

strongly upregulated in Gln-deprived macrophages (van der Vos *et al*, 2012; Palmieri *et al*, 2017; Shang *et al*, 2020). These results suggest that the increased consumption of Gln by Gln-addicted cancer cells creates a state of Gln shortage in the tumor microenvironment (TME) that might trigger the acquisition of a GS-high protumoral phenotype of TAMs. However, evidence with this respect is elusive.

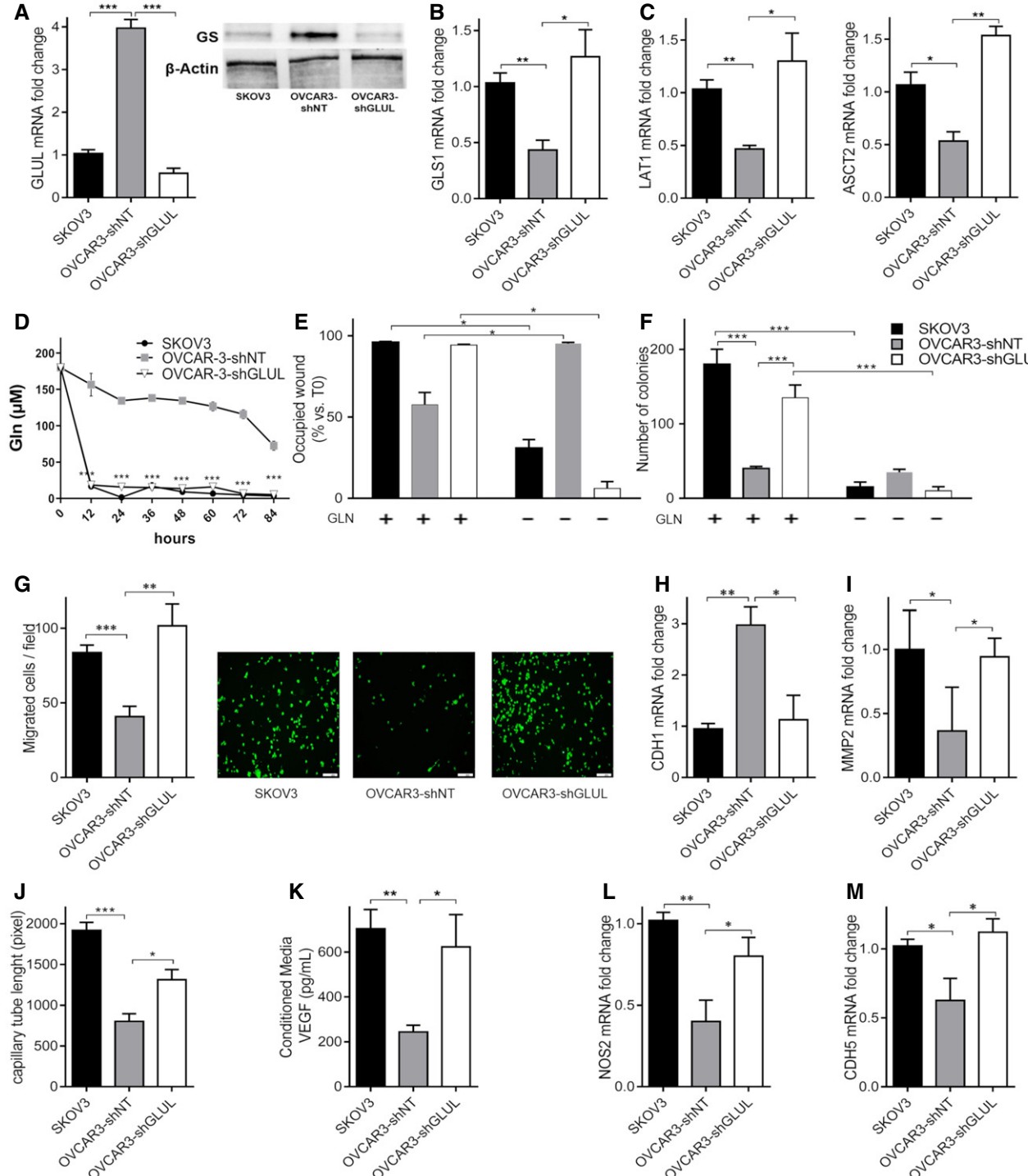

**Figure 1.**

    

**Figure 1. GS knockdown in OVCAR3 cells reprograms glutamine metabolism.**

A  qRT–PCR quantification of *GLUL* mRNA transcript and GS protein expression in SKOV3, OVCAR3-shNT, and OVCAR3-sh*GLUL* after 24 h of culture (*n* = 3 biological replicates).

B, C  qRT–PCR quantification of (B) glutaminase (*GLS1*) and (C) glutamine transporters (*LAT1* and *ASCT2*) mRNA in SKOV3, OVCAR3-shNT, and OVCAR3-sh*GLUL* after 24 h of culture (*n* = 3 biological replicates).

D  Extracellular Gln levels measured at different indicated time points in SKOV3, OVCAR3-shNT, and OVCAR3-sh*GLUL* cells (*n* = 3 biological replicates).

E, F  Effect of 24 h of glutamine deprivation on cell proliferation measured by occupied wound assay (E) (*n* = 6 biological replicates) and clonogenic test (F) (*n* = 3 biological replicates) in SKOV3, OVCAR3-shNT, and OVCAR3-sh*GLUL* cells.

G  Cell migration propensity was measured through a Matrigel-coated micropore filter in SKOV3, OVCAR3-shNT, and OVCAR3-sh*GLUL* (*n* = 3 biological replicates). The migration of the cells, stained with Calcein-AM, was analyzed by fluorescence microscopy. The scale bars indicate 100 μm.

H, I  Cell invasion valuated by qRT–PCR quantification of (H) *CDH1* and (I) *MMP2* transcript levels in SKOV3, OVCAR3-shNT, and OVCAR3-sh*GLUL* (*n* = 3 biological replicates).

J  Quantitative analysis of the total lengths of the endothelial capillary network formed by HUVEC cells cocultured with SKOV3, OVCAR3-shNT, and OVCAR3-sh*GLUL* for 6 h (*n* = 6 biological replicates).

K  Quantification of released VEGF by SKOV3, OVCAR3-shNT, and OVCAR3-sh*GLUL* cells in conditioned media (*n* = 3 biological replicates).

L, M  Angiogenesis propensity measured by qRT–PCR quantification of (L) *NOS2* and (M) *CDH5* mRNA levels in SKOV3, OVCAR3-shNT, and OVCAR3-sh*GLUL* cells (*n* = 3 biological replicates).

Data information: Data are displayed as mean ± SEM. Statistical significance was calculated by one-way ANOVA analyses with Tukey correction (A–C, G–M), two-way ANOVA analyses with Tukey correction (D–F) and defined as *$P < 0.05$, **$P < 0.01$, ***$P < 0.001$.

Here, we provide unprecedented evidence of a metabolic crosstalk mechanism occurring between glutaminolytic ovarian cancer cells and macrophages, which is confirmed in ovarian cancer patients and we unravel the protumoral role of a known metabolite.

## Results

Gln dependency of low-aggressive OVCAR3 cells was achieved through GS (*GLUL*) stable silencing, as assessed at both the RNA and protein levels (Fig 1A). Additionally, in this study, we also included SKOV3 cells as they are *per se* highly glutaminolytic and express very low levels of GS (Fig 1A). Compared to their scramble control (shNT), targeting *GLUL* in OVCAR3 cells leads to increased *GLS1* transcript levels (Fig 1B), upregulated cellular glutamine transporters, *LAT1* and *ASCT2* (Fig 1C) and Gln uptake (Fig 1D), to a similar extent as observed in SKOV3 cells. *In vitro*, Gln was crucial for OVCAR3-sh*GLUL* cell proliferation (Fig 1E and F) compared with OVCAR3-shNT cells. The propensity for migration (Fig 1G), invasion (Fig 1H and I), and angiogenesis (Fig 1J–M) in OVCAR3-sh*GLUL* and, similarly, in SKOV3 cells was also enhanced in a Gln dependent fashion compared with OVCAR3-shNT cells, indicating that Gln addiction, generated in OVCAR3 cells by GS downmodulation, correlates with the acquisition of invasive, metastatic, and angiogenetic features of OVCAR3 cells.

We then cocultured LPS/IFNγ-stimulated human macrophages with OVCAR3-sh*GLUL* or OVCAR3 cells and tested their polarization status. A marked decrease in M1 markers (Fig 2A) and increase in M2 markers (Fig 2B) were evident in LPS/IFNγ macrophages cocultured with OVCAR3-sh*GLUL* (and SKOV3) but not in those cocultured with OVCAR3 cells, compared with LPS/IFNγ macrophages alone. This was paralleled by augmented GS expression at both RNA and protein levels (Fig 2C and D), as well as higher glutamine transporters (Fig 2E) but lower *GLS1* expression (Fig 2F) in LPS/IFNγ-stimulated macrophages cocultured with SKOV3 or OVCAR3-sh*GLUL* compared with OVCAR3-shNT cells, suggesting that Gln addiction in OVCAR3 cells positively correlates with the induction of a high GS, M2-like phenotype in macrophages. OVCAR3-sh*GLUL* and SKOV3 media displayed higher release of IL-10 (Fig 3A), which

is known to induce GS expression (Palmieri *et al*, 2017) in macrophages, and of NAA (although to a lower extent and at a later time point in SKOV3) compared with OVCAR3 media (Fig 3B), suggesting the OVCAR3-sh*GLUL* cells might instate a complex crosstalk mechanism by releasing different signals in the TME. From a metabolic point of view, increased extracellular NAA concentration correlated with higher asparagine (Asn) uptake (Fig 3C) and higher intracellular levels of citrate, aspartate (Asp), and glutamate (Glu) (Fig 3D–F) in OVCAR3-sh*GLUL* compared with OVCAR3-shNT and SKOV3 cells. At a later time point (84 h), SKOV3 cells also displayed higher citrate and aspartate (Fig EV1A and B), associated with a mild but significant increase in NAA. A significant decrease in NAA levels (Fig EV1C) is induced by GS overexpression in SKOV3 cells (Fig EV1D). Both *N*-acetyl transferase (*NAT8L*) and ATP-citrate lyase (*ACLY*) were upregulated in OVCAR3-sh*GLUL* and SKOV3 compared with OVCAR3 cells (Fig 3G and H), suggesting that citrate might represent a source of both Acetyl-CoA and Asp, the latter via oxaloacetate transamination, which is facilitated by Glu accumulation (Fig 3I).

Then, we wondered whether NAA, together with IL-10, contributes to polarize macrophages toward an "M2-like" phenotype. NAA treatment in Gln-deprived LPS/IFNγ (Fig 4A and B) macrophages acted similarly to IL-10 in promoting an M2-like state, since *CD206* and *CD163* mRNA levels were significantly higher in NAA-treated compared with LPS/IFNγ macrophages, but lower than NAA/IL-10-treated cells (Fig 4A and B). Conversely, co-treatment of NAA and NAA/IL-10 cells with an excess of the antagonist *N*-methyl-ᴅ-aspartate (NMDA, 30 times more than NAA) completely rescued the M1 to M2 switch, since it prevented *CD206* and *CD163* mRNA increase (Fig 4A and B) while promoting *CD80* mRNA levels and TNFα release (Fig 4C and D). These results were confirmed by FACS studies (Fig EV2), indicating that NAA possibly acts by competing with ligands of the NMDA receptor and not through its metabolism. In support to this, the signaling effect of NAA in LPS/IFNγ macrophages was not abolished by silencing aspartoacylase (*ASPA*), the enzyme that catalyzes NAA deacetylation (Figs 4E and EV3A) when localized in the cytosol (Hershfield *et al*, 2006). Furthermore, following NAA treatment (10 μM), the extracellular levels of the molecule were unchanged after 24 h (Fig 4F).

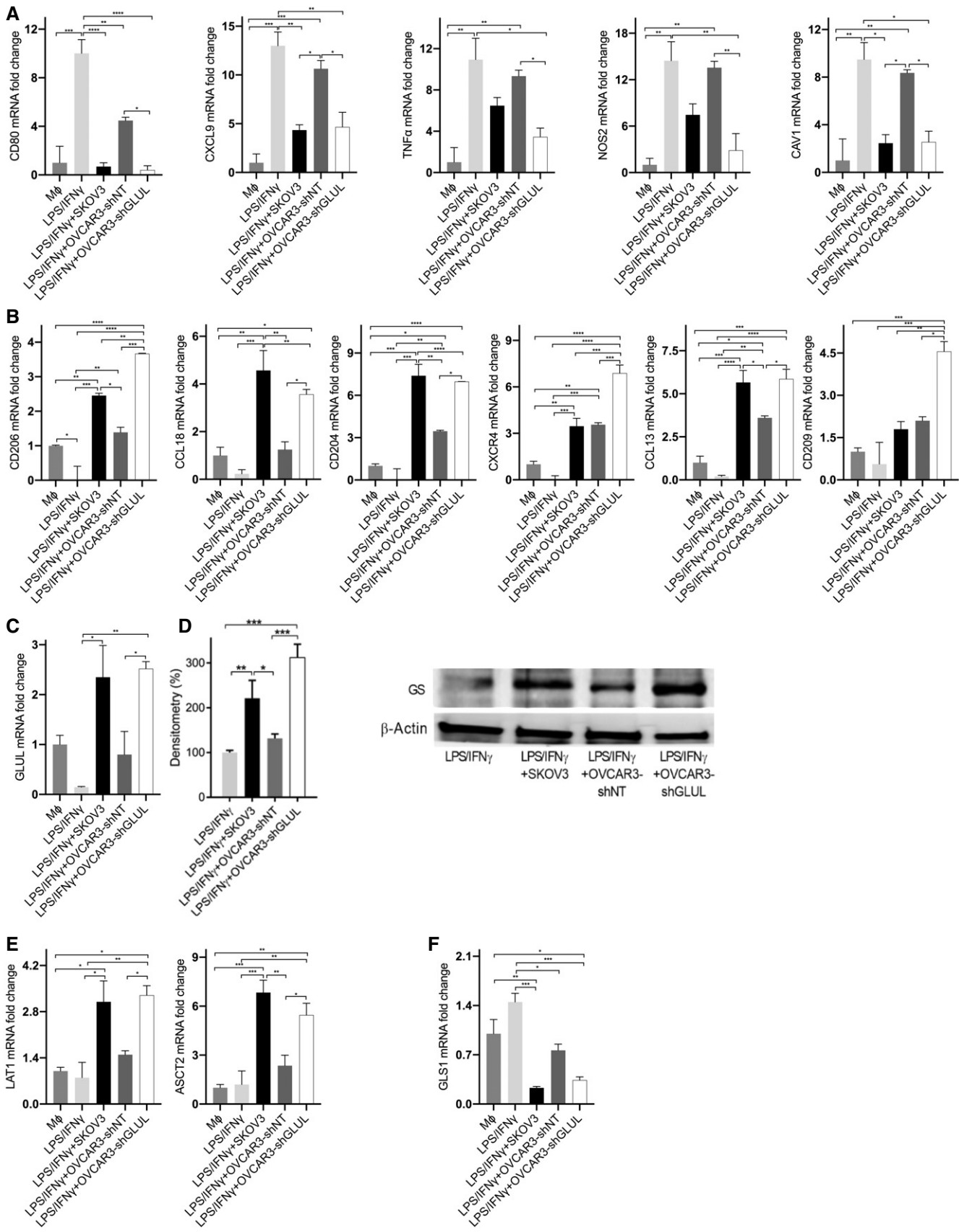

**Figure 2.**

◀ **Figure 2. OVCAR3-sh*GLUL* cells enhance LPS/IFNγ macrophage polarization toward a M2-like, GS-high phenotype compared with OVCAR3 cells.**

A, B qRT–PCR quantification of (A) M1 and (B) M2 markers in resting (Mφ) and LPS/IFNγ macrophages after a 24 h of coculture with or without SKOV3, OVCAR3-shNT, or OVCAR3-sh*GLUL* cells (n = 3 biological replicates).

C qRT–PCR quantification of *GLUL* transcript in resting (Mφ) and LPS/IFNγ macrophages after coculture with or without SKOV3, OVCAR3-shNT, or OVCAR3-sh*GLUL* cells (n = 3 biological replicates).

D GS protein expression quantification (with representative blot) in LPS/IFNγ macrophages after coculture with or without SKOV3, OVCAR3-shNT, or OVCAR3-sh*GLUL* cells (n = 3 biological replicates).

E, F qRT–PCR quantification of (E) *LAT1* and *ASCT2* and (F) *GLS1* transcript level in resting (Mφ) and LPS/IFNγ macrophages after coculture with or without SKOV3, OVCAR3-shNT, or OVCAR3-sh*GLUL* cells (n = 3 biological replicates).

Data information: Data are displayed as mean ± SEM. For all panels, statistical significance was calculated by one-way ANOVA analyses with Tukey correction and defined as *P < 0.05, **P < 0.01, ***P < 0.001, ****P < 0.0001.

Mirroring these data, a 48-h time course incubation with 10 μM or even 20 μM NAA did not result in significant changes in its intracellular content (Fig 4G). The M1 to M2 switch promoted by NAA was associated with an increase in GS levels, which were completely abrogated by NMDA (Fig 4H). NAA-mediated trans-differentiation in Gln-rich medium (Fig EV3B–E) induced a M2-like switch with a similar trend to that observed when Gln was absent, although the increase in M2 markers was significantly less pronounced compared with the corresponding Gln-depleted samples, particularly for the treatments with NAA, IL-10, and IL-10/NAA (Fig EV3F–H). On the contrary, the increase in *CD80* was less pronounced in the Gln-depleted compared with the Gln-rich samples (Fig EV3I). Furthermore, NAA treatment on IL-10 differentiating macrophages enhanced the M2-like phenotype, as indicated by the increased *CD206*, *CD163*, and *GLUL* levels in NAA/IL-10 compared with IL-10 macrophages (Fig EV3J–L). Altogether, these results indicate that NAA acts together with IL-10 in inducing an M2-like phenotype in macrophages by impeding NMDA receptor activation, which consequently leads to GS expression, previously shown to sustain the M2-like polarization (Palmieri et al, 2017; Menga et al, 2020).

To obtain deeper insights into the mechanism by which NAA acts as NMDAR antagonist, we analyzed the available NMDAR structures crystallized in complex with agonists and antagonists. The list of the PDB entries highlighted through by pGenThreader, used for the following analysis, is reported in Dataset EV1. The cryo-EM structure of *Homo sapiens* NMDAR is available as apo-protein under the code 6irg.pdb (Zhang et al, 2018). *Xenopus laevis* cryo-EM/crystallized structures used for our comparative analysis were solved in complex with Glu and Gly agonists (as reported in 5iov.pdb, (Zhu et al, 2016) and 5uow.pdb, (Lü et al, 2017)) or with allosteric inhibitor (Ro25-6981, i.e., as reported in 5iov.pdb and 4tll.pdb, (Lee et al, 2014) or with the MK-801 channel blocker (5uow.pdb, (Lü et al, 2017)). In addition, the *R. norvegicus* GluN2D ligand-binding core (3oem.pdb) was solved in complex with NMDA co-agonist (as reported in 3oem.pdb (Vance et al, 2011)). NMDA (from 3oem.pdb) and Glu (from 5uow.pdb) ligands appear well superimposed within the human GluN2 ligand-binding core of the human NMDAR (6irg.pdb), and NMDA-binding residues (i.e., residues within 4 Å from NMDA) are a subgroup of residues also involved in direct interactions with the Glu agonist (Fig EV4A–D). The docked NAA, showing an acetyl group in place of the methyl group of NMDA, interacts with most of residues interacting with NMDA agonist, but also with H485, known for being involved in direct interactions with Glu, indicating the NAA is an effective ligand of NMDAR. H485 plays a crucial role in NMDAR conformational changes (Ieong Tou et al, 2014), and NAA

competes with Glu in the Glu agonist-binding region at GluN2 subunit, thus modulating the $Ca^{2+}$ influx (North et al, 2015). Based on this prediction, we evaluated at a single-cell level the $Ca^{2+}$ influx triggered by NAA, compared to NMDA, in resting and LPS/IFNγ macrophages (Fig 4I). A significant $Ca^{2+}$ influx was recorded only in 10% of the NAA-stimulated LPS/IFNγ macrophages, whereas the addition of NMDA induced $Ca^{2+}$ influx in 60% of the cells. In resting macrophages, both NAA and NMDA did not induce any significant change in fluorescence. These results suggest that, at variance with NMDA, NAA structure does not elicit $Ca^{2+}$ influx associated with NMDAR activation.

Moving to *in vivo*, the transcriptome of 307 TCGA ovarian cancer tissues was analyzed to estimate tumor purity and the infiltration level of M2 macrophages, assuming that gene expression levels are the weighted sum of the expression levels from the cell population mix constituting the cancer tissue (Li et al, 2020). The analysis showed that *GLUL* expression level negatively correlates with tumor purity (Fig 5A), suggesting that tumor cells are not a major contributor to *GLUL* expression variations and, more importantly, that cells other than tumor cells contribute to *GLUL* expression. Further, *GLUL* expression level positively correlates with the infiltration level of M2 macrophages (Fig 5B), suggesting that the latter are main contributors to *GLUL* expression level. In support of this statement, application of a single-cell signature matrix from GSE146026 RNA sequencing data (Izar et al, 2020) to the same TCGA dataset revealed in grade 3 patients a strong upregulation of *GLUL* expression in TAMs compared with cancer cells (Fig 5C). Similarly, but in other datasets than TCGA, grade 3 ovarian cancer samples clustering for high levels of M2 macrophage markers (*CD163* and *MRC1*, the latter encoding for CD206 protein) also showed high *GLUL* expression level (Fig 5D), further supporting the idea that M2 macrophages are mostly contributing to *GLUL* expression. Further, high *GLUL* expression levels related to high infiltration of M2 macrophages corresponded to low cumulative survival rate of patients (Fig 5E).

Inspired by previous *in vitro* data, we performed a correlation analysis for three genes involved in glutamine metabolism (*GLS1*, and the two glutamine transporters *SLC7A5* and *SLC1A5*) and *NAT8L*. We found these genes co-expressed with *MKI67*, i.e., a marker gene of cellular proliferation (Fig 5F), indicating that Gln metabolism and transport across cells, and NAA synthesis directly or indirectly mark an active proliferative state. Furthermore, *GLS1* and *ACLY* were significantly co-expressed with *NAT8L*, linking glutamine and citrate metabolism to NAA synthesis (Fig 5F). To validate the *in silico* observations, we immunostained the ovarian tissue

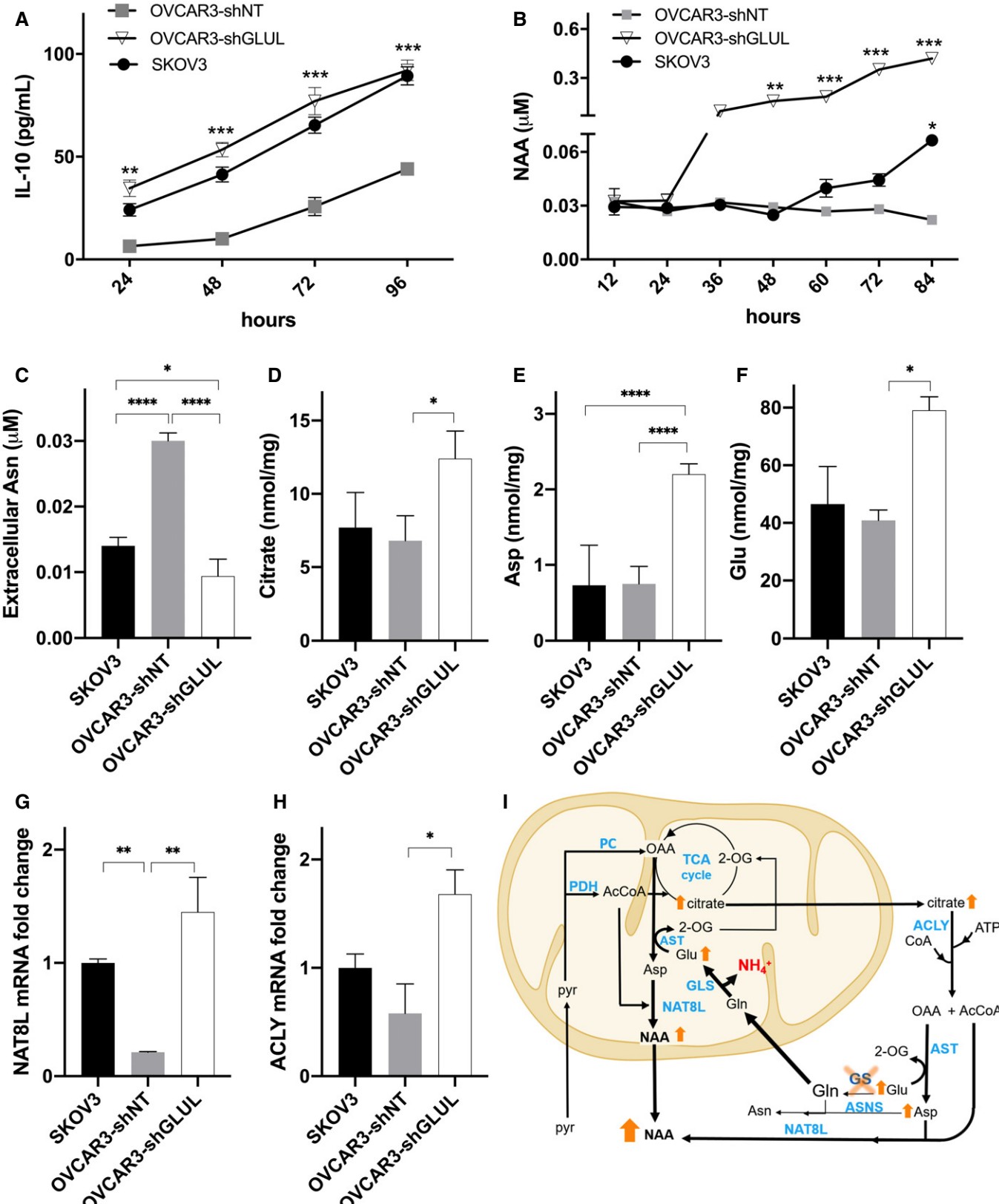

**Figure 3.**

**Figure 3.** *GLUL* knockdown effect on release of IL-10 and NAA and on consequent metabolic aspects.

A, B    Extracellular levels of (A) IL-10, and (B) NAA in SKOV3, OVCAR3-shNT, and OVCAR3-sh*GLUL* cells (*n* = 3 biological replicates).
C–F    Extracellular levels of Asn (C) and cytosolic levels of citrate (D), Asp (E), and Glu (F) in SKOV3, OVCAR3-shNT, and OVCAR3-sh*GLUL* cells (*n* = 3 biological replicates).
G, H    qRT–PCR quantification of (G) *NAT8L* and (H) *ACLY* mRNA in SKOV3, OVCAR3-shNT, and OVCAR3-sh*GLUL* cells (*n* = 3 biological replicates).
I    Ammonia production driven by glutaminolysis interferes with mitochondrial dehydrogenases, favoring citrate, AcCoA and OAA mitochondria accumulation. Up-regulation of *NAT8L* along with Glu accumulation, driving OAA transamination to Asp, leads to NAA production. NAA is also synthesized by *NAT8L* in the cytoplasm using AcCoA derived from citrate via *ACLY*. In the absence of intracellularly synthesized Gln, Glu drives OAA transamination leading to Asp, which accumulates due to decreased ASNS activity. AcCoA: Acetyl-CoA; ASNS: asparagine synthetase; AST: aspartate transaminase; *ACLY*: ATP-citrate synthase; *GLS1*: glutaminase; GS: glutamine synthetase; NAA: *N*-acetylaspartate; *NAT8L*: *N*-acetyltransferase 8 like; OAA: oxaloacetate; 2-OG: 2-oxoglutarate; PC: pyruvate carboxylase; PDH: pyruvate dehydrogenase.

Data information: Data are displayed as mean ± SEM. Statistical significance was calculated by two-way ANOVA analyses with Tukey correction (A, B), one-way ANOVA analyses with Tukey correction (C–H), and defined as *$P < 0.05$, **$P < 0.01$, ***$P < 0.001$, ****$P < 0.0001$.

microarrays (TMAs) for *GLS1*, Ki67, GS, and *CD163* and carried out immunofluorescence (IF) microscopy. We observed a significantly positive stage-dependent correlation between *CD163*$^+$GS$^+$ macrophage infiltration and glutaminolytic proliferating (Ki67$^+$*GLS1*$^+$) cells (Figs 5G–I and EV5C and D).

To further confirm these findings, we isolated macrophages from ascitic fluids of ovarian cancer patients, clustered in stages according to the FIGO guidelines. As expected, the expression of the M2 markers *CD206* and *CD163* correlated with the stage, as their mRNA level in TAMs is higher in more advanced tumor samples (Fig 6A). Surprisingly, *GLUL* levels (Fig 6B), but not *ASPA* levels (Fig EV5A) in ascitic macrophages also correlated with the stage (Fig 6B), indicating the GS represents a specific feature of protumoral macrophages in ovarian cancer. To establish the existence of the crosstalk mechanism identified in the *in vitro* setting, we quantified IL-10 and NAA levels in the same ascitic fluids. In accordance with previous findings, IL-10 (Fig 6C) in ascitic fluid was augmented at the increase in the cancer stage and this correlates with *GLUL* levels in TAMs (Fig 6D). Similarly, NAA levels in ascitic fluid also augmented (Fig 6E) with the cancer stage and it correlated with *GLUL* levels in TAMs (Fig 6F). On the contrary, Gln levels did not follow the same trend (Fig EV5B). The high *GLUL* expression in ascite-associated macrophages, the glutaminolytic, and *NAT8L* expressing phenotype of ascitic cancer cells and their strongly positive correlation were also confirmed by analyzing RNA-Seq data (E-MTAB-4162, E-MTAB-5498, Worzfeld *et al*, 2018) (Fig EV6A–E).

Altogether, these *in vivo* findings support the presence of this crosstalk mechanism.

## Discussion

The role of Gln addiction in ovarian cancer cells has been already unraveled, as it associates with increased invasion and aggressiveness (Yang *et al*, 2014). Furthermore, it might be triggered/exacerbated by lactate, through upregulation of Gln transport and catabolism (Pérez-Escuredo *et al*, 2016). However, how Gln addiction impacts on the TME is far from being elucidated. Here, we suggest that cancer cells exploit their addiction for Gln in order to establish a crosstalk mechanism in the TME which supports their malignant, pro-invasive phenotype through the engagement of protumoral TAMs. Based on our previous studies, we initially hypothesized that ovarian cancer cells might establish a pressure on TAMs by consuming extracellular Gln, leading to an M2-like phenotype associated with higher *GLUL*, which is known to respond to a condition of extracellular Gln shortage (Palmieri *et al*, 2017; Mazzone *et al*, 2018). Unexpectedly, with this approach we identified superimposing mechanisms in which glutaminolytic ovarian cancer cells enforce a macrophage switch toward a GS-expressing protumoral phenotype. Surprisingly, NAA was found as a major player in this crosstalk, displaying an "IL-10-like" effect. NAA is the second most abundant amino acid derivative in the brain obtained by AcCoA-mediated acetylation of aspartate. NAA is a source of acetate necessary for myelin synthesis although it is also involved in other tasks, such as neuronal osmoregulation, and axon-glial signaling (Moffett *et al*, 2007). It is also known to accumulate in many cancer types (Stadlbauer *et al*, 2012), and its increase correlates with poor survival (Zand *et al*, 2016). Furthermore, it has been identified as one of the most highly upregulated metabolites in ovarian cancer (Zand *et al*, 2016). Based on the finding that ASPA expression does not correlate with tumor NAA (Zand *et al*, 2016), a non-catabolic role of NAA has been hypothesized (Bogner-Strauss, 2017).

Here, we demonstrate that acquired glutaminolysis in ovarian cancer cells, which has been already associated with ovarian cancer invasiveness, links to NAA production. Strong evidence in cancer studies supports the role of glutaminolysis in driving aspartate synthesis for NAA production (Lou *et al*, 2016; Wynn *et al*, 2016). Additionally, conditions increasing GS expression are reported to reduce NAA synthesis (Wynn *et al*, 2016). This evidence is in line with our results on the metabolic rewiring following GLUL ablation in OVCAR3 cells. It is conceivable to hypothesize that ammonia accumulation in *GLUL*-deficient cells might lower TCA cycle dehydrogenases flux (Ott *et al*, 2005; Ott & Vilstrup, 2014; Drews *et al*, 2020), leading to citrate accumulation, as found in our model (Fig 3I). In the mitochondria, oxaloacetate might be rerouted to transamination, facilitated by high Glu levels, leading to Asp, which is promptly acetylated by *NAT8L*. In the cytosol, *ACLY* upregulation might promote oxaloacetate synthesis from citrate and subsequent Asp production, which is then acetylated. This interpretation is strongly supported by our findings that both NAT8L gene expression and ACLY gene expression are strongly induced in GLUL-deficient cells. Our data point to the metabolic rewiring linked to high *GLS1/GLUL* ratio as a main contributor to NAA synthesis. Indeed, genetically unrelated SKOV3 cells, which are highly glutaminolytic but retain a low but detectable GS protein level (Fig 1A), release NAA at high times. In the same time interval, *GLUL*-overexpressing SKOV3 cells strongly reduce their ability to produce NAA compared with SKOV3 cells (Fig EV1C).

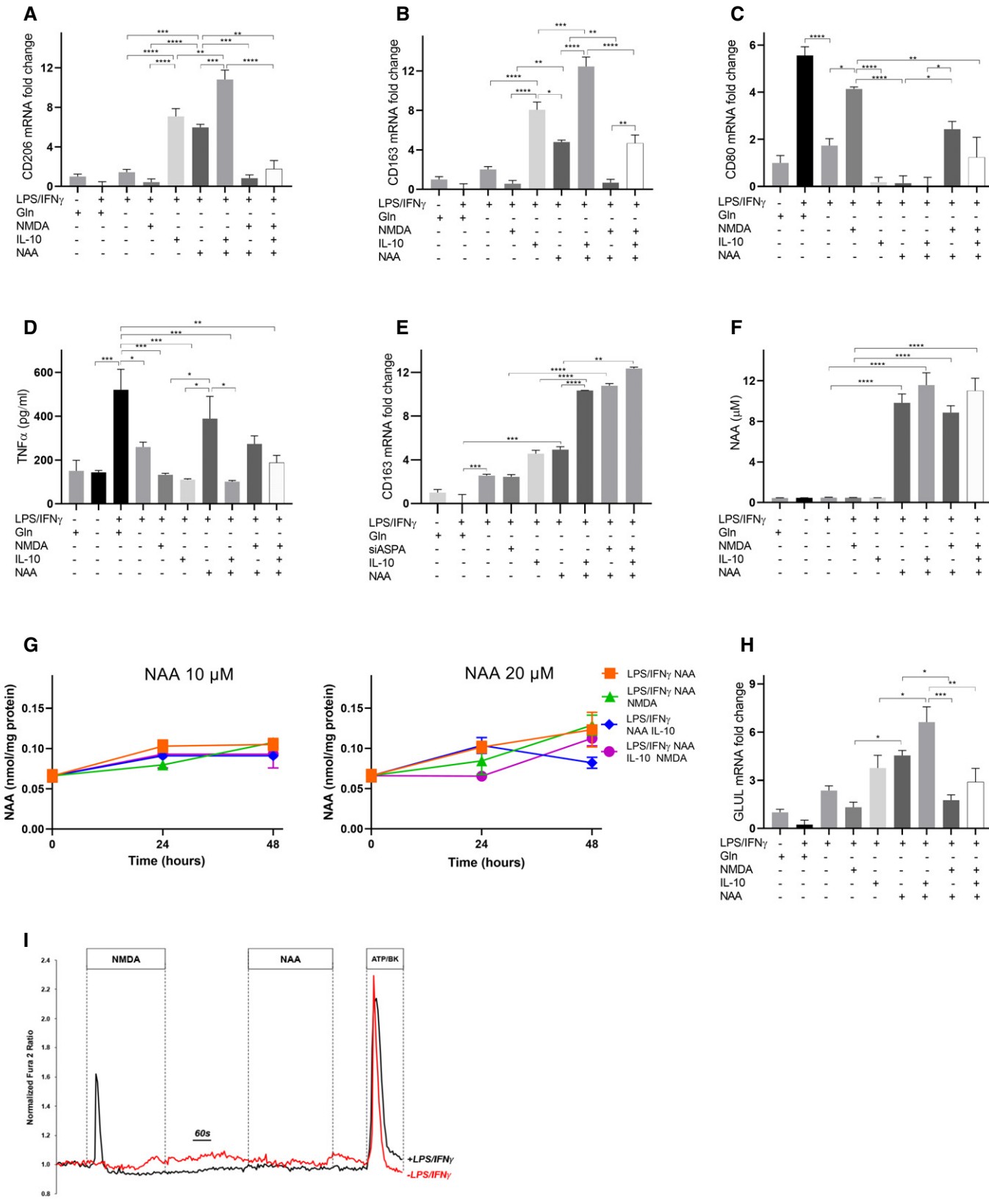

**Figure 4.**

**Figure 4.  Effect of NAA on macrophage polarization.**

A–C  qRT–PCR quantification of (A) *CD206*, (B) *CD163*, and (C) *CD80* mRNA in LPS/IFNγ macrophages treated with NAA (10 μM) and/or IL-10 and/or NMDA in a Gln-depleted medium (*n* = 3 biological replicates).

D    Extracellular TNFα levels in LPS/IFNγ macrophages treated with NAA (10 μM) and/or IL-10 and/or NMDA in a Gln-depleted medium (*n* = 3 biological replicates).

E    qRT–PCR quantification of *CD163* mRNA in *ASPA*-silenced LPS/IFNγ macrophages treated with NAA (10 μM) and/or IL-10 and/or NMDA in a Gln-depleted medium (*n* = 3 biological replicates).

F    Extracellular NAA levels in LPS/IFNγ macrophages treated for 24 h with NAA (10 μM) and/or IL-10 and/or NMDA in a Gln-depleted medium (*n* = 3 biological replicates).

G    Intracellular NAA levels in LPS/IFNγ macrophages treated for 24 h with NAA (10 and 20 μM) and/or IL-10 and/or NMDA in a Gln-depleted medium (*n* = 3 biological replicates).

H    qRT–PCR quantification of *GLUL* mRNA in LPS/IFNγ macrophages treated with NAA (10 μM) and/or IL-10 and/or NMDA in a Gln-depleted medium (*n* = 3 biological replicates).

I    Representative traces of $Ca^{2+}$ influx in resting (red line) and LPS/IFNγ (black line) macrophages, measured after NMDA (300 μm), NAA (10 μM), or ATP (100 μM)/Bradykinin (1 μM) treatments in a Gln-depleted medium.

Data information: Data are displayed as mean ± SEM (A–C, E–I) or ± SD (D). Statistical significance was calculated by one-way ANOVA analyses with Tukey correction (A–H) and defined as $*P < 0.05$, $**P < 0.01$, $***P < 0.001$, $****P < 0.0001$.

Our findings clearly identify an unprecedented crosstalk mechanism between cancer cells and macrophages involving NAA, which acts synergically with IL-10 and Gln deprivation to polarize macrophages toward a high-GS, "M2-like" phenotype. Our *in vivo* data support the existence of this crosstalk in ovarian cancer patients, in which we found that *GLUL* levels from ascitic fluid macrophages were increased proportionally according to the cancer stage and significantly correlated with the amount of both IL-10 and NAA, measured in the ascitic fluid. Additionally, immunohistochemical data from a large set of ovarian cancer tissues confirm that macrophages are the main contributor to *GLUL* in high-grade tumors.

Our data suggest that the mechanism of NAA action is not linked to its metabolic processing in macrophages, but rather to its function as NMDA receptor (NMDAR) ligand. Indeed, concomitant treatment with NMDA abrogates its effect on LPS/IFNγ macrophages. NMDARs are tetrameric ion channels hosting two glutamate ionotropic receptor NMDA type subunits 1 (GluN1, encoded by *GRIN1*) and 2 (GluN2, encoded by *GRIN2*), whose activation requires concurrent binding of glycine and L-glutamate to the GluN1 and GluN2 subunits, respectively (Erreger *et al*, 2007). Four distinct GluN2 subunits (GluN2A-GluN2D) with different tissue-specific expression patterns may participate in the formation of the tetrameric ion-channel. While GluN2A and GluN2B appear to be mainly expressed in brain, GluN2C and GluN2D appear to be more ubiquitous with GluN2D that appears to be the most expressed paralog in ovary (Glasgow *et al*, 2015). For investigating the putative role played by NAA in NMDAR function, we analyzed interactions of the cited agonists and inhibitors by superimposing the human NMDAR structure to *X laevis* and *R. norvegicus* structures, allowing to dock Glu, Gly, and NMDA agonists, as well as Ro25-6981 allosteric inhibitor and the MK-801 channel blocker, in the corresponding/superimposed binding regions of the human NMDAR (Fig EV4). The docking of NAA in the human NMDAR structure shows that NAA binds to the Glu agonist-binding region similarly to NMDA. In addition, NAA establishes with its acetyl moiety-specific binding interactions with the residue H485, typically involved in interactions with the Glu agonist (Fig EV4). This docking analysis sustains the hypothesis that NAA affinity for GluN2 subunit is very similar to the ones displayed by Glu and higher than the ones displayed by NMDA agonists for filling Glu-binding region. Thus, NAA can bind NMDAR more efficiently than NMDA and with an efficiency very similar to

the ones observed for Glu, due to the presence of the acetyl-moiety in place of the methyl group on the aspartate amino group. On this regard, it is known that the introduction of a methyl group on the amine moiety of Asp and Glu ligands produces a loss of affinity of the two *N*-methyl derivatives for Glu agonist-binding region, as observed by EC50 estimations performed in the presence of the cited Glu, Asp, and their *N*-methyl derivatives (Erreger *et al*, 2007). However, this is counterbalanced by the specific interactions between the acetyl-moiety and H485, which plays a key role in the NMDAR conformational changes necessary for $Ca^{2+}$ influx (Delano, 2002; Erreger *et al*, 2007; Lobley *et al*, 2009; Vance *et al*, 2011; Bossis *et al*, 2014; Ieong Tou *et al*, 2014; Lee *et al*, 2014; North *et al*, 2015; Zhu *et al*, 2016; Lü *et al*, 2017; Chauhan *et al*, 2018; Zhang *et al*, 2018). It is retained that the binding of NAA might be stronger than the NMDA binding due to the presence of stronger interactions induced by the presence of the acetyl-moiety. By comparing residues involved in the binding of Glu and NAA, it is observed that both ligands bind to the same binding pocket residues. At variance with NMDA, both Glu and NAA show short-range (below 3.5 Å) direct binding interaction with H485, but only Glu shows short-range binding interactions with Y730. Both residues synergically play a key role in conformational changes at GluN2 subunit after Glu binding (Ieong Tou *et al*, 2014). On the contrary, shorter aspartate and NMDA cannot establish direct interactions with none of them and are therefore less efficient agonists compared with Glu (Erreger *et al*, 2007). These observations suggest that NAA direct interactions with H485 confer to NAA a greater affinity for GluN2 subunit agonist-binding domain, compared with other structurally related agonists (i.e., NMDA or Asp). However, based on our docking analyses, NAA establishes H-bond interactions with H485, but not with Y730, at variance with what observed for Glu. The different sets of interactions displayed by NAA might determine a sort of asymmetry in the GluN2-binding region at variance with Glu. This asymmetry might be responsible for the acquisition of a more rigid GluN2 subunit conformational changes, which might result in the impairment of $Ca^{2+}$ influx. In support of this hypothesis, NAA did not elicit any $Ca^{2+}$ uptake in M1-like macrophages, at variance with NMDA. These findings, together with our data supporting lack of NAA influx and catabolism by *ASPA* in macrophages, suggest a functional rewiring role of the molecule on macrophages through NMDAR binding. Incidentally, NAA at concentration lower than

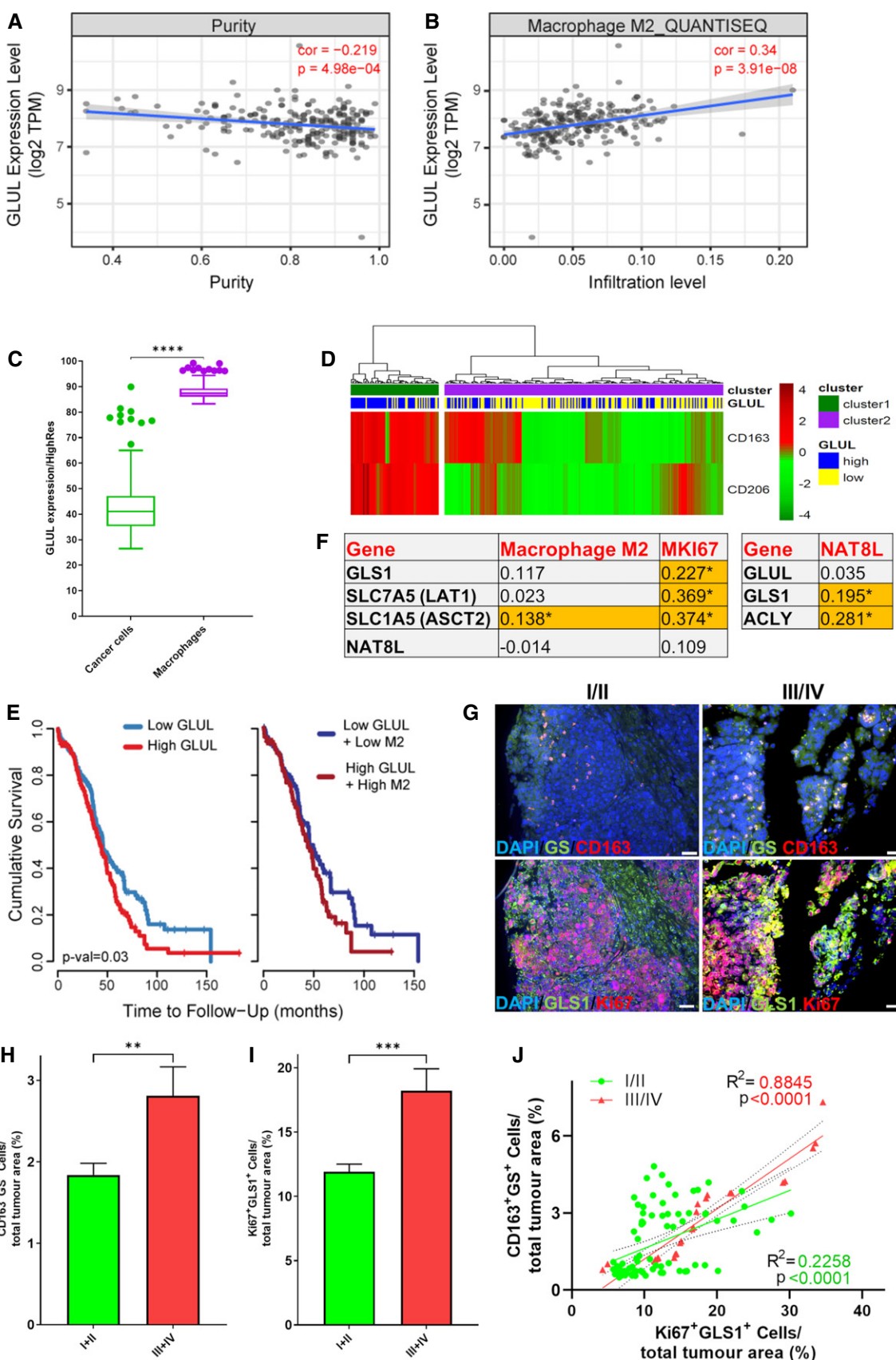

Figure 5.

◀

3 mM is known to bind to NMDAR without inducing $Ca^{2+}$ influx (Rubin *et al*, 1995). Furthermore, NMDAR antagonism skews human macrophages toward a M2-like phenotype (Nowak *et al*, 2019). Our data indicate that NAA release contributes to the rewiring of proinflammatory macrophages toward protumoral functions as a NMDAR ligand, structurally related to Glu or Asp, but with no effect on $Ca^{2+}$ influx. We believe that, in the presence of IL-10, the polarizing action of NAA goes through competitive binding to NMDAR but this does not exclude that, in other contexts, *ASPA* can affect macrophage functions, both dependently and independently of its enzymatic activity. Overall, our work supports the idea to explore NAA as a predictive biomarker for disease progression in ovarian cancer. It also corroborates with human data our previous findings on the pharmacological inhibition of GS as anti-tumor strategy but particularly it suggests for the first time that antagonistic stimulation of NMDA receptors might offer a therapeutic asset to re-educate TAMs toward an anti-tumor, immunostimulatory phenotype.

## Materials and Methods

### Cell culture

Ovarian cancer cell lines OVCAR3 and SKOV3 were purchased from ATCC®. Cells were grown in RPMI 1640 from Gibco (31870074, Thermo Fisher Scientific) supplemented with 10% fetal bovine serum (FBS), 2 mM L-glutamine, 100 IU/ml penicillin, and 100 μg/ml streptomycin. When starved, cells were transferred into RPMI medium without glutamine supplemented with dialyzed FBS. *GLUL* KO or overexpressing cells were generated using lentiviral particles (TL312740V; LVP015) from AMSBIO® and selected by puromycin (10 μg/ml) or blasticidin (10 μg/ml), respectively. Transduction of SKOV3 cells with lentiviral particles for *GLUL* overexpression was monitored via the RFP fluorescent signal under CELENA® S Digital Imaging System (Logos Biosystems, South Korea).

Human monocytes were obtained from healthy blood donor buffy coats. Ascitic macrophages were obtained from ascitic fluids of ovarian cancer patients under an institutional review board-approved protocol (University of Bari, Prot. 0070295). Informed consent was obtained for all subjects participating to the study. The experiments were conformed to the principles set out in the WMA Declaration of Helsinki and the Department of Health and Human Services Belmont Report. Ascitic macrophages and human monocytes from buffy coats were isolated as reported (Adhikary *et al*, 2017; Palmieri *et al*, 2017). Briefly, buffy coats or ascitic fluid diluted 1:1 with isolation buffer (PBS $Ca^{2+}/Mg^{2+}$ free added of 1 mM EDTA) was very gently layered on top of a density gradient medium (Lymphoprep™) and centrifuged at 1,200 *g* for 20 min at room temperature. The cloudy ring containing PBMCs was collected in a new centrifuge tube, added with 3× volume of cold isolation buffer, and centrifuged at 250 × *g* for 12 min at 4°C. The supernatant was removed gently. The PBMC pellet was resuspended in cold PBS $Ca^{2+}/Mg^{2+}$ free containing 2 mM EDTA and 0.5% BSA and then added with CD14 MicroBeads (Miltenyi Biotec Inc.) as described previously (Palmieri *et al*, 2015). After an incubation of 15 min at 4°C, the mix was centrifuged at 300 *g* for 10 min at 4°C and the pellet resuspended in 1 ml of PBS/EDTA/BSA and then put on LC columns for the positive selection of $CD14^+$ cells. The labeled monocytes (from buffy coats) were collected in a new tube, resuspended in RPMI 1640 supplemented with 2 mM L-glutamine, 100 IU/ml penicillin, and 100 μg/ml streptomycin (Gibco, Invitrogen), and incubated at 37°C under a humid atmosphere with 5% $CO_2$. Growth factors (rh GM-CSF or rh M-CSF) were added to address differentiation to the appropriate phenotype for six days. Cell viability was verified in the cultures by Trypan blue exclusion. After differentiation, macrophages were stimulated with LPS/IFNγ (for M1 polarization) or IL-10 (for M2 polarization). Treatments were performed in a trans-differentiation setting, consisting of a polarization with LPS/IFNγ for 24 h, then a wash out, followed by Transwell cocultures with cancer cells or by treatment for 24 h with 10 or 20 μM NAA,

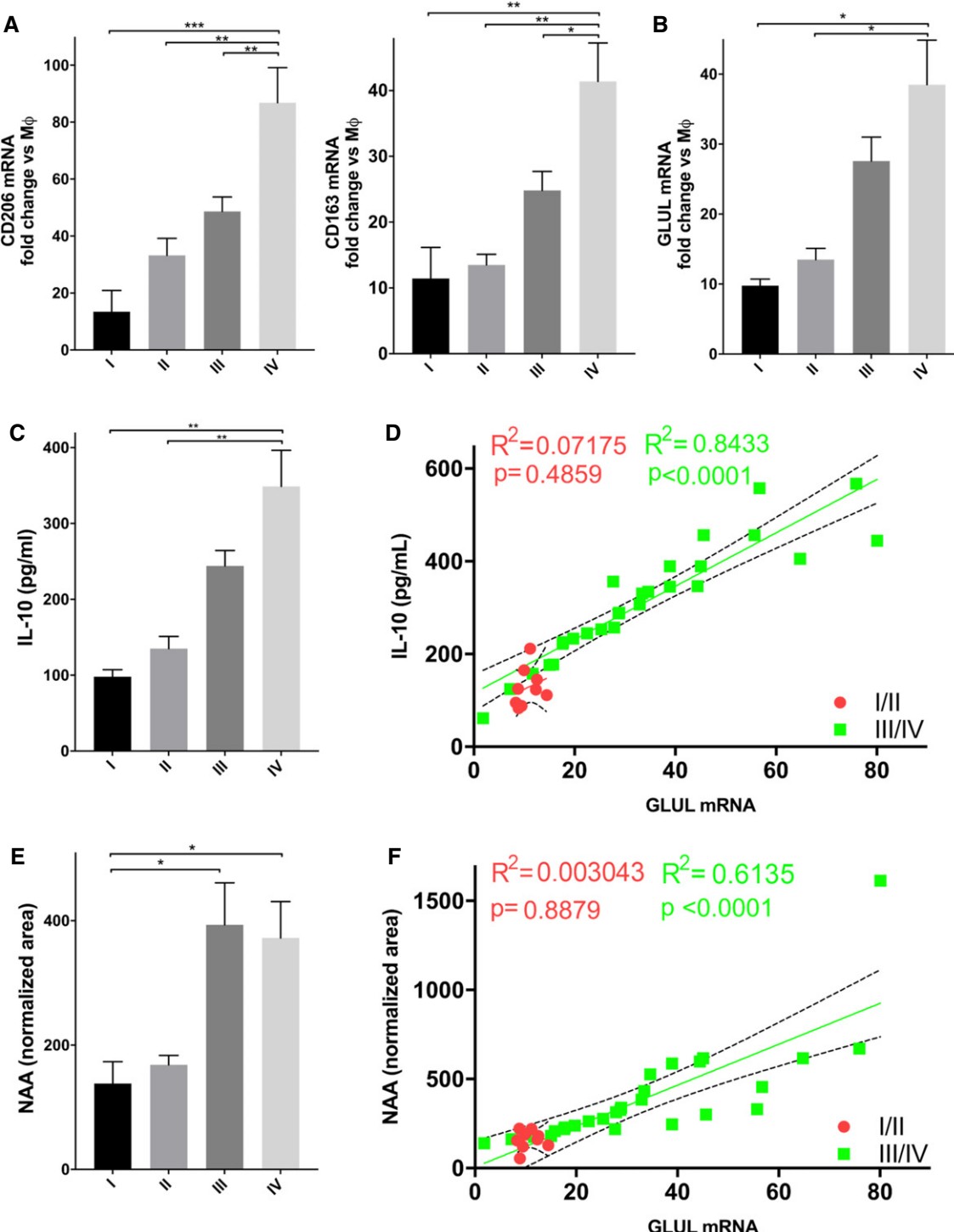

**Figure 6.** *GLUL* is expressed by TAMs in ascitic fluid of ovarian cancer patients, and it correlates with NAA and tumor progression.

A, B    qRT–PCR quantification of (A) *CD206* and *CD163*, and (B) *GLUL* mRNA in TAMs isolated from ascitic fluid derived from ovarian cancer patients (*n* = 53, 5 of stage I; 7 of stage II; 28 of stage III; 13 of stage IV). Control samples are represented by human resting macrophages (Mφ).

C, D    IL-10 levels (C) in the ascitic fluid isolated from ovarian cancer patients (*n* = 38, 4 of stage I; 7 of stage II; 20 of stage III; 7 of stage IV) and (D) correlation with *GLUL* levels. R: Pearson correlation coefficients; $R^2$: means "the goodness of fit". Dashed lines indicate 95% confidence intervals.

E, F    NAA levels (E) in the ascitic fluid isolated from ovarian cancer patients (*n* = 38, 4 of stage I; 7 of stage II; 20 of stage III; 7 of stage IV) and (F) correlation with *GLUL* levels. R: Pearson correlation coefficients; $R^2$: means "the goodness of fit". Dashed lines indicate 95% confidence intervals.

Data information: Data are displayed as mean ± SEM. For (A–C) panels, statistical significance was calculated by one-way ANOVA analyses with Dunn's correction, for (C, E) Mann–Whitney test, for (D, F) by Pearson correlation coefficients and defined as *$P < 0.05$, **$P < 0.01$, ***$P < 0.001$.

10 ng/ml IL-10, 300 μM NMDA in Gln rich or depleted medium (Kaindl *et al*, 2012; Li *et al*, 2016).

For experiments in the M2 differentiation setting, Mφ macrophages were treated with 10 μM NAA and/or 300 μM NMDA and then 10 ng/ml IL-10 in glutamine-depleted medium.

Cell viability was determined as described (Menga *et al*, 2017; Palmieri *et al*, 2017). CD14$^+$ cells isolated from ascitic fluid were counted and mRNA extracted for qRT–PCR studies.

### Wound healing and colony assay

Wound healing and clonogenic assays were performed as indicated (Yang *et al*, 2014).

### Tube formation

Matrigel Basement Membrane Matrix (BD Biosciences) was diluted with EBM-2 medium and coated in 24-well plates at 37°C for 1 h. Then, $5 \times 10^4$ HUVECs were seeded alone or cocultured with an equivalent number of OVCA cells in the EBM-2 medium on Matrigel. Cocultured OVCA cells were seeded into Transwells and incubated in the same well with HUVECs. The tube formation ability of HUVECs was measured at 6 h with or without OVCA cells. After incubation, the total length of the tubular structures was quantified.

### Matrigel invasion assay

Cell invasion assay was performed for 8 h in Boyden chamber coated with 20 μl of reduced growth factor Matrigel (1:6 dilution; BD Biosciences) followed by Calcein-AM (Life Technologies) staining (4 μg/ml). After incubation, medium in the lower chamber was aspirated, and invaded cells were treated with 5% glutaraldehyde in PBS for 15 min to fix the cells and then washed in PBS solution three times. The noninvaded cells on the inner surface of upper chambers were carefully removed by using a cotton swab. Finally, invaded cells were counted under 20× magnification for at least three fields per insert (Yang *et al*, 2014).

### Flow cytometry

The expression of CD80 and MHCII on macrophage surfaces was studied by flow cytometry. Purified cells following specific treatments were washed with FACS buffer and $3 \times 10^5$ cells labeled with fluorescence-labeled anti-human antibody (PE anti-CD80, BioLegend; Alexa Fluor®594 anti-HLA-DR, R&D Systems) for 30 min at 4°C in dark as per manufacturer's recommendation. After labeling, cells were washed with PBS, resuspended in 300 μl of the same saline solution until acquisition which was performed using a BD FACSCelesta™ flow cytometer. Data analysis was performed with Kaluza software (Beckman Coulter). Suitable negative isotype controls were used to rule out the background fluorescence for the used fluorochromes.

### Intracellular Ca$^{2+}$ measurements

For cytosolic Ca$^{2+}$ recordings, resting macrophages were seeded on 15-mm Ø glass coverslips, treated with LPS/INFγ or left under basal condition. After 24 h, cells were loaded with 6 μM Fura-2-AM (Thermo Fisher Scientific, Waltham, MA, USA) for 30 min at 37°C in RPMI, followed by 15 min in an extracellular solution to allow Fura-2 de-esterification. The coverslips with dye-loaded cells were mounted in an open perfusion chamber (RC-42LP) and placed in a Quick Exchange Platform (both from Warner Instruments, Holliston, MA, USA) that allows perfusion and temperature control. During the experiments, the cells were perfused (at 2 ml/min) with a bath solution containing (in mM): 140 NaCl, 5 KCl, 1 MgCl$_2$, 10 HEPES, 1 CaCl$_2$, 5 glucose; pH 7.4. Bath Ringer's solution was modified whenever needed adding different compounds depending on the experimental approach. Recordings were carried out using an inverted Eclipse TE2000-S microscope (Nikon, Shinagawa, Tokyo, Japan) equipped for single-cell fluorescence evaluations and imaging analysis. Each Fura-2-AM loaded sample was illuminated every 5 s through a 40× oil immersion objective (numerical aperture = 1.30) at 340 and 380 nm. The emitted fluorescence was passed through a dichroic mirror, filtered at 510 nm (Omega Optical, Brattleboro, VT, USA), and captured by a cooled CCD CoolSNAP HQ camera (Photometrics, Tucson, AZ, USA). Fluorescence measurements were carried out following stimulation with NMDA 300 μm, NAA 10 μM, or ATP 100 μM/Bradykinin 1 μM (as a positive control), using the MetaFluor Fluorescence Ratio Imaging Software (version 7.7.3.0, Molecular Devices, San Jose, CA, USA). The ratio of the fluorescence signal acquired upon excitation at 340 and 380 nm was normalized to the basal fluorescence ratio obtained in the absence of the stimulus.

### Analyses of TCGA and GEO human ovarian cancer datasets

Clinical data, gene expression, tumor purity, and M2 macrophage infiltration levels for 307 epithelial ovarian cancer samples reported in The Cancer Genome Atlas (TCGA, https://www.cancer.gov/tcga) were recovered and analyzed using TIMER2.0 (Li *et al*, 2020). In particular, quanTIseq (Finotello *et al*, 2019) was selected for estimating M2 macrophage infiltration level and Spearman's rho value was calculated and tested in correlation analyses. Cumulative survival probability in patients with the highest or lowest 50% *GLUL* expression (and M2 macrophage infiltration level) was evaluated using Kaplan–Meier curves and log-rank comparison.

Raw data were also gathered from four ovarian cancer GEO datasets (https://www.ncbi.nlm.nih.gov/gds/, GSE6008, GSE14764, GSE23554, GSE26712) selecting cases according to high-grade parameter. All samples profiled by HG133A Affymetrix array were processed by RMA Express v1.2.0 (Robust Multi-Array Average), and expression values were log$_2$-transformed. Samples were divided into two categories (highest and lowest 50% *GLUL* expression). Hierarchical clustering analysis of samples (heatmap) was conducted on *CD163* and *CD206* expression levels using Euclidean distance and Ward.D2 cluster aggregation method within R (v 3.6.1) package "pheatmap".

High Resolution CIBERSORTx algorithm (Newman *et al*, 2019) was applied to obtain expression profiles from each microenvironment cell as well as from tumor entity. A signature matrix was built from GSE146026 (Izar *et al*, 2020) single-cell RNA sequencing data which included tumor, macrophages, and fibroblast cells. Then, cell-specific expression profiles were imputed from TGCA (Bell *et al*, 2011) data including 380 ovarian cancer samples using FPKM normalization. Finally, *GLUL* expression levels were extracted from each cell type and compared among them.

RNA sequencing (RNA-Seq) data for ascitic tumor cells (TU) and TAM-matched samples from HGSOC patients (stage 3) are deposited at EBI ArrayExpress (accession numbers E-MTAB-4162 and E-MTAB-5498). Transcripts per million (TPM) values were used as measure of mRNA expression.

### Human tissue samples

Tissue microarrays (TMA) containing 70 (OV701) and 99 (OV991) ovarian cancer samples were purchased from US Biomax. These materials were commercially available anonymized and de-identified. Since the tissues were commercially purchased, the study has been exempted from requiring ethical approval. Paraffin-embedded tissue slides were baked overnight, then de-waxed, and rehydrated according to standard protocols. Tissue sections were then subjected to antigen retrieval in TRIS-EDTA at 95°C for 20 min (pH = 9). The solution is then cooled for an additional 20 min on the benchtop prior to buffer rinse. Tissues were then incubated overnight with rabbit anti-glutaminase (Abcam, Cat# ab156876), mouse anti-GS (Sigma MAB302), mouse anti-Ki67 (Novacastra, NCL-L-Ki67-MM1), and rabbit anti-CD163 (Abcam, ab87099) in a humid chamber in the dark at room temperature. Appropriate secondary antibodies Alexa 488 and 594-conjugated (Invitrogen) were used. The slides were then mounted using nucleus-specific DAPI counterstain and coverslipped. Slides were then imaged using fluorescence microscopy on a Nikon Eclipse 80i-ViCO.

### Other methods

RNA isolation, real-time PCRs, and Western blotting analysis were performed as described before (Lauderback *et al*, 2003; Palmieri *et al*, 2014, 2015; Costiniti *et al*, 2018). *ASPA* silencing was obtained as described (Prokesch *et al*, 2016). Metabolite quantification was achieved with LC-MS/MS analysis as reported (Palmieri *et al*, 2015, 2017). Briefly, cells were lysed in methanol:water (80:20). For extracellular quantification, 100 µl of medium was extracted with 100% methanol. After solvent evaporation, pellets were resuspended in deionized water. Liquid chromatography tandem mass spectrometry (LC-MS/MS) analysis was performed with a Quattro Premier mass spectrometer interfaced with an Acquity UPLC system (Waters). Calibration curves were established using standards, processed under the same conditions as the samples, at five concentrations. The best fit was determined using regression analysis of the peak analyte area. The multiple reaction monitoring transitions selected in the negative ion mode were $m/z$ 190.95 > 110.89 for citrate and 174.00 > 88.00 for NAA, in the positive ion mode were $m/z$ 147.20 > 83.90 for Gln, $m/z$ 148.20 > 83.90 for Glu, and $m/z$ 134,16 > 73.86 for Asp. Chromatographic resolution was achieved using HSS T3 (2.1 × 100 mm, 1.8-µm particle size, Waters) for citrate and BEH C18 (2.1 × 50 mm, 1.7-µm particle size, Interchim) for amino acids. For all columns, the flow rate was 0.3 ml/min. IL-10 and VEGF (Thermo Scientific) and TNFα (R&D Systems) were quantified by ELISA.

### Statistical analysis

Data are shown as means ± SEM or ± SD. At least two independent experiments were performed. The statistical significance of comparisons between two groups was determined using two-tailed (paired or unpaired) *t*-test and Mann–Whitney test for cell cultures and patients. For comparison among more than two groups, ANOVA test was applied and followed by *post hoc* Tukey test, or ANOVA followed by Dunn's correction. For Timer2.0 (http://timer.cistrome.org/) analysis, Spearman's test implemented in the R cor.test function was applied. For survival curve analysis, Cox proportional hazard model test was applied. For analysis on the GEO datasets, *P*-value among categorical variables was calculated by Fisher's exact test. For human tissue sample analysis, statistical significance was calculated by Pearson correlation coefficients. Data were considered significant for *$P < 0.05$, **$P < 0.01$, ***$P < 0.001$, ****$P < 0.0001$.

## Data availability

This study includes no data deposited in external repositories.

**Expanded View** for this article is available online.

### Acknowledgements

MM is supported by an ERC Consolidator Grant (ImmunoFit, #773208), FWO (G0D1717N and G066515N), and StichtingTegenKanker (2014-197). AC is supported by grant from the Italian Ministry of Economic Development (MISE) (F/200076/01-02/X45). GC is supported by AIRC (FIRC-AIRC 25254, AM is supported by EMBO (ASTF 610-2015), and FEBS Fellowship grants.

### Author contributions

AM, MM, and ACas conceptualized the project; AM, MF, IS, and ACas contributed to data curation; MM and ACas contributed to funding acquisition; AM, MF, and IS contributed to investigation; MF, IS, MCV, RG, ADG, LL, ACam, AG, GC, MC, and CLP contributed to methodology; ACas contributed to supervision. VL and GC contributed to resources; AM, MM and ACas wrote the original draft of the manuscript; AM, MF, IS, MCV, RG, MM, and ACas reviewed and edited the manuscript. All authors read and approved the manuscript.

### Conflict of interest

The authors declare that they have no conflict of interest.

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
