## [Review Process File · EMBO Reports]

N-acetylaspartate release by glutaminolytic ovarian cancer cells sustains pro-tumoral macrophages

Alessio Menga, Maria Favia, Iolanda Spera, Maria Carmela Vegliante, rosanna gissi, Anna De Grassi, Luna Laera, Annalisa Campanella, Andrea Gerbino, Giovanna Carrà, Marcella Canton, Vera Loizzi, Ciro Pierri, Gennaro Cormio, Massimiliano Mazzone, and Alessandra Castegna

DOI: [10.15252/embr.202051981](https://doi.org/10.15252/embr.202051981)

Corresponding author(s): Alessandra Castegna (alessandra.castegna@uniba.it), Alessandra Castegna (alessandra.castegna@uniba.it), Massimiliano Mazzone (massimiliano.mazzone@kuleuven.vib.be)

Review Timeline:

Submission Date:	5th Nov 20
Editorial Decision:	15th Dec 20
Revision Received:	17th Apr 21
Editorial Decision:	21st May 21
Revision Received:	10th Jun 21
Accepted:	21st Jun 21

Editor: Deniz Senyilmaz Tiebe

Transaction Report:

Dear Dr. Castegna,

Thank you for submitting your manuscript to EMBO Reports. Three referees agreed to review your manuscript. So far, we have received two referee reports that are copied below. Given the comments of these referees, I would like to ask you to begin revising your study along the lines suggested by the referees.

Please note that this is a preliminary decision made in the interest of time, and that it is subject to change should the third referee offer very strong and convincing reasons for this. As soon as/if we receive the final report on your manuscript, we will forward it to you as well.

We concur with the referees that the proposed NAA mediated crosstalk between the glutamine addicted ovarian cancer cells and the tumor associated macrophages is in principle very interesting. However, the referees also raise significant concerns that need to be fully addressed to consider publication here.

- The proposed link between glutamine addiction of tumor cells, NAA production and M2 polarization is not sufficiently supported by the data (referee #3, paragraph that starts as 'The data provided using...').
- The proposed regulation of macrophage polarization by NAA via NMDAR requires additional support (referee #3, paragraphs that start as 'In my opinion the possibility...' and 'Lastly, the authors suggest..').
- Overall, the provided experimental controls are insufficient, methods need to be further clarified (both referees).

I find the reports informed and constructive, and believe that addressing the concerns raised will significantly strengthen the manuscript.

Should you be able to address all criticisms in full, we could consider a revised manuscript. I do realize that addressing all the referees' criticisms will require a lot of additional time and effort and be technically challenging. I would therefore understand if you wish to publish the manuscript rapidly and without any significant changes elsewhere, in which case please let us know so we can withdraw it from our system.

If you decide to thoroughly revise the manuscript for EMBO Reports, please revise your manuscript with the understanding that the referee concerns (as in their reports) must be fully addressed and their suggestions taken on board. Please address all referee concerns in a complete point-by-point response. Acceptance of the manuscript will depend on a positive outcome of a second round of review. It is EMBO reports policy to allow a single round of revision only and acceptance or rejection of the manuscript will therefore depend on the completeness of your responses included in the next, final version of the manuscript.

We generally allow three months as standard revision time. As a matter of policy, competing manuscripts published during this period will not negatively impact on our assessment of the conceptual advance presented by your study. However, we request that you contact the editor as soon as possible upon publication of any related work, to discuss how to proceed. Should you

foresee a problem in meeting this three-month deadline, please let us know in advance and we may be able to grant an extension.

*** Temporary update to EMBO Press scooping protection policy:

We are aware that many laboratories cannot function at full efficiency during the current COVID-19/SARS-CoV-2 pandemic and have therefore extended our 'scooping protection policy' to cover the period required for a full revision to address the experimental issues highlighted in the editorial decision letter. Please contact the scientific editor handling your manuscript to discuss a revision plan should you need additional time, and also if you see a paper with related content published elsewhere.***

1. A data availability section providing access to data deposited in public databases is missing (where applicable).
2. Your manuscript contains statistics and error bars based on $n=2$. Please use scatter plots in these cases.

Supplementary/additional data: The Expanded View format, which will be displayed in the main HTML of the paper in a collapsible format, has replaced the Supplementary information. You can submit up to 5 images as Expanded View. Please follow the nomenclature Figure EV1, Figure EV2 etc. The figure legend for these should be included in the main manuscript document file in a section called Expanded View Figure Legends after the main Figure Legends section. Additional Supplementary material should be supplied as a single pdf labeled Appendix. The Appendix includes a table of content on the first page with page numbers, all figures and their legends. Please follow the nomenclature Appendix Figure Sx throughout the text and also label the figures according to this nomenclature. For more details please refer to our guide to authors.

Please note that for all articles published beginning 1 July 2020, the EMBO Reports reference style will change to the Harvard style for all article types. Details and examples are provided at <https://www.embopress.org/page/journal/14693178/authorguide#referencesformat>

2) individual production quality figure files as .eps, .tif, .jpg (one file per figure).

3) a .docx formatted letter INCLUDING the reviewers' reports and your detailed point-by-point responses to their comments. As part of the EMBO Press transparent editorial process, the point-by-point response is part of the Review Process File (RPF), which will be published alongside your paper. For more details on our Transparent Editorial Process, please visit our website:

<https://www.embopress.org/page/journal/14693178/authorguide#transparentprocess>

4) a complete author checklist, which you can download from our author guidelines (<<http://embor.embopress.org/authorguide>>). Please insert information in the checklist that is also reflected in the manuscript. The completed author checklist will also be part of the RPF.

5) Please note that all corresponding authors are required to supply an ORCID ID for their name upon submission of a revised manuscript (<<https://orcid.org/>>). Please find instructions on how to link your ORCID ID to your account in our manuscript tracking system in our Author guidelines (<<http://embor.embopress.org/authorguide>>).

6) We replaced Supplementary Information with Expanded View (EV) Figures and Tables that are collapsible/expandable online. A maximum of 5 EV Figures can be typeset. EV Figures should be cited as 'Figure EV1, Figure EV2' etc... in the text and their respective legends should be included in the main text after the legends of regular figures.

- For the figures that you do NOT wish to display as Expanded View figures, they should be bundled together with their legends in a single PDF file called *Appendix*, which should start with a short Table of Content. Appendix figures should be referred to in the main text as: "Appendix Figure S1, Appendix Figure S2" etc. See detailed instructions regarding expanded view here: <<http://embor.embopress.org/authorguide#expandedview>>.

7) We would also encourage you to include the source data for figure panels that show essential data.

Numerical data should be provided as individual .xls or .csv files (including a tab describing the data). For blots or microscopy, uncropped images should be submitted (using a zip archive if multiple images need to be supplied for one panel). Additional information on source data and instruction on how to label the files are available <<http://embor.embopress.org/authorguide#sourcedata>>.

8) Our journal encourages inclusion of *data citations in the reference list* to directly cite datasets that were re-used and obtained from public databases. Data citations in the article text are distinct from normal bibliographical citations and should directly link to the database records from which the data can be accessed. In the main text, data citations are formatted as follows: "Data ref: Smith et al, 2001" or "Data ref: NCBI Sequence Read Archive PRJNA342805, 2017". In the Reference list, data citations must be labeled with "[DATASET]". A data reference must provide the database name, accession number/identifiers and a resolvable link to the landing page from which the data can be accessed at the end of the reference. Further instructions are available at <<http://embor.embopress.org/authorguide#datacitation>>.

9) Please make sure to include a Data Availability Section before submitting your revision - if it is not applicable, make a statement that no data were deposited in a public database. Primary datasets (and computer code, where appropriate) produced in this study need to be deposited in an appropriate public database (see <<http://embor.embopress.org/authorguide#dataavailability>>).

The accession numbers and database should be listed in a formal "Data Availability " section (placed after Materials & Method) that follows the model below. Please note that the Data Availability Section is restricted to new primary data that are part of this study.

Data availability

10) Regarding data quantification, please ensure to specify the name of the statistical test used to generate error bars and P values, the number (n) of independent experiments underlying each data point (not replicate measures of one sample), and the test used to calculate p-values in each figure legend. Discussion of statistical methodology can be reported in the materials and methods section, but figure legends should contain a basic description of n, P and the test applied. Please note that error bars and statistical comparisons may only be applied to data obtained from at least three independent biological replicates. Please also include scale bars in all microscopy images.

I look forward to seeing a revised version of your manuscript when it is ready. Please let me know if you have questions or comments regarding the revision.

Yours sincerely,

Deniz Senyilmaz Tiebe

Deniz Senyilmaz Tiebe, PhD
Editor
EMBO Reports

Referee #2:

Metabolic communication between tumor and stromal cells emerge as a key player on modulating activation and functions of immune cells and suppressive nature of the tumor microenvironment. In this manuscript, Menga et. al. reported that glutamine addiction in ovarian tumor cells provides metabolic cues to instruct macrophage polarisation in coordination with IL10 and become

glutamine-producing macrophages. Overall, this work stems from an early work published by same groups in Cell Reports and provide an interesting link on glutamine addition in tumor cells with this macrophage phenotype. However, there are several issues should be addressed to strengthen the conclusion and techniques used in this work.

1. The rationale on using IFN γ /LPS in polarizing macrophages is not clear. Do the authors test IL4/IL13 treatment in the same setting? This is critical for linking their findings with in vivo observation in the tumor microenvironment since IL4/IL13 is known to affect macrophage phenotypes in the tumor microenvironment.

2. The analyses performed for figure 4a and 4b was not clarified in a sufficient details in the manuscript. In my opinion, the correlations on figure 4a and 4b are marginal.

3. Figure 4h and I look promising. However, the correlation and p value for figure 4i should be provided. Similarly, Figure 4j and k should also be presented with those information. This is of particular importance since the authors group stage 1 and 2 vs stage 3 and 4. Is there strong correlation along with stage of ovarian cancer?

4. Extended from the previous point, it will also important to examine GLUL expression in tumors of EOC patients used in Figure 4. This will strengthen whether the expression of GLUL in tumor negatively associates with IL10 expression and GLUL expression in macrophages.

Referee #3:

Here Menga et al. propose an interesting model whereby glutamine addicted tumors deplete Gln and produce an interesting metabolite, NAA. This NAA acts not as a metabolite but as an antagonist for the NMDA receptor and drives the development of so called "M2" macrophages. They report higher expression of CD163 and CD206 by macrophages from ascites fluid of high grade tumors and their analysis of TCGA data shows higher NAA levels and higher markers of "M2" cells in higher grade tumors. I find this model to be intriguing but there seem to be substantial gaps in the support for the model.

The data provided using the ovarian cell lines shows that glutamine addiction results in substantial depletion of glutamine in the cultures. The co-culture figures are consistent with the depletion of glutamine by the tumor cells during co-culture and the known effects of Gln deprivation on macrophages. However, the NAA production by the tumor lines only occurs in one of the lines that depletes glutamine, but not the other. Thus the negative correlation between the ability to make Gln and the ability to deplete it from the culture fits well, and the tumor functional assays fit with the availability of Gln but not the production of NAA. In vivo Gln is very abundant in blood making depletion unlikely and macrophage "M2" phenotypes, they argue, correlate with Gln availability and GS expression, but these do not correlate with NAA production by the tumor. I understand the argument that high grade tumors have high NAA and that may correlate with increased expression of "M2" phenotypes but the data in Figure 3 are all generated in the absence of Gln. If I am not misunderstanding the data, this seems to be a disconnect. It may be true that NAA drives an M2 like phenotype in the absence of Gln and that Gln deprivation drives an M2 phenotype and that Gln deprivation drives aggressiveness but as I see it the data don't fully support a connection between these findings. Moreover, if as the authors suggest, M2 macrophages are "the main contributors to GLUL expression level variations" in tumors then what is the connection to the first figures of this paper? In addition, Figure 4I and 4K seem to suggest substantial variability in GLUL

expression across tumors. This apparent 80 fold spectrum together with the different levels of GLUL expressed in the two lines assessed here do not seem concordant with the conclusion that GLUL variation in the tumors is only due to M2 macrophages. Moreover, although I am not familiar with the analysis in 4B it would seem there is only about a 4 fold spectrum of "infiltration" so I do not understand how this could lead to an 80 difference in GLUL expression.

In my opinion the possibility that NAA alters macrophage phenotype is very interesting but the proposed mechanism, as an antagonist of NMDAR, is not particularly well supported. In figure 3, the NMDA agonist and inhibitor seem to have tremendous effects on their own (especially in the promotion of CD80 expression). Are there endogenous, constitutively expressed ligands in this system? There are no data regarding expression of the receptor and no demonstration of the expected biochemistry of the receptor when NMDA is added. The cited papers provide little comfort. KIM et al (2017) shows a different NMDA antagonist (agmatine) is neuroprotective but the reference also says it can block nos so it is not clear it is working via the receptor. Li et al (2016) shows that THP1 cells treated with PMA apparently can express the NMDAR and that MK801 suppresses expression of the "M1" like cytokine TNF. The latter supports the story here but because this is such an interesting idea, I feel the details of the receptor must be well documented here.

A third major weakness is the consistent lack of controls in the macrophage experiments. The phenotypes of the beginning populations are not shown making several figures hard to fully interpret. For example Figure 2 and Figure 3C-H are all stimulated, there are no resting controls to compare. The expectation is that the levels of CD206 etc. are extraordinarily low in LPS and lfn stimulated cells making 10 fold increases of potentially negligible relevance. The resting levels must be shown in these experiments and in the co-culture experiments as well. Moreover, the key markers here are easily assessed on the cell surface by FACS yet the data are nearly all qPCR and there are no ELISAs for the chemokines.

Lastly, the authors suggest that rather than working as a metabolite that the NAA is a receptor antagonist. Despite this, when they knockdown ASPA there are substantial increases in CD163 mRNA. Although the levels of ASPA are never shown and the efficacy of the knockdown is not shown, the authors conclude "the mechanism of NAA action is not linked to its metabolic processing".

There are several additional minor concerns that lessen enthusiasm or raise questions regarding the mechanism:

The methods are lacking in several key details. How are the macrophages made? Palmieri 2017 is cited, but it refers to Palmieri 2015 which then cites Infantino 2011 which seems to be a review. Le et al refers to PMA/LPS/IFN but that is for THP1. Similarly, how are the ascites-derived macrophages isolated and handled. The data refer to values relative to M0. What does that mean? It is assumed that the NAA is very acidic. Acidic conditions have been shown to favor alternative activation of macrophages. Has this been ruled out?

What is the glutamine level in the ascites?

As noted NAA levels and expression of Nat8l has already been associated with poor outcomes by Zand et al. thus the statement "Our work paves the way towards the possibility to explore NAA as a novel, predictive biomarker for disease progression in ovarian cancer." Is not really appropriate.

Figure 1F should show the effects of supplementation of Gln. Given the growth differences it could be lactate etc.

Figure 3 also should show re-introduction of glutamine.

What is the status of ASPA and Nat8l in the macs and tumors?

Figure 3C shows only a slight increase of CD206 with NAA+IL10 compared to these agents alone. This is not even additive, much less synergistic as suggested. The effect on CD163 is perhaps additive.

There are multiple references to "M2 skewing" and "polarizing to toward 'M2-like'" and "M1 to M2 switch" but as noted above the controls are not shown. Is this really just interfering with establishment of M1?

Figure 4A is not convincing.

Contrary to the contentions of these authors there are multiple animal models of ovarian cancer. In particular, because the proposed model involves modification of macrophage phenotypes, information could be derived from xenografts of these human lines in to immunocompromised mice. These sorts of studies are not necessarily needed here but paucity of models should not be suggested.

Different gene sets are used for different figures for M1 and M2. CD206 should be shown in figure 2.

In figure 3 it is not clear why the IL10 and MK108 combo was tested (with or without NMDA agonist)? This is superfluous data that doesn't add to the experiment and makes a confusing chart even more confusing.

EMBO REPORTS 2020-5198

“Release of N-acetylaspartate by glutaminolytic ovarian cancer cells sustains the pro-tumoral phenotype of macrophages in synergy with IL-10”

Response to reviewers

Referee #2:

The rationale on using IFN γ /LPS in polarizing macrophages is not clear. Do the authors test IL4/IL13 treatment in the same setting? This is critical for linking their findings with *in vivo* observation in the tumor microenvironment since IL4/IL13 is known to affect macrophage phenotypes in the tumor microenvironment.

We chose the LPS/IFN γ because this phenotype is strongly *in vitro* primed to kill cancer cells. This choice in our opinion further strengthens the role of NAA in macrophage rewiring, since the molecule is able to reprogram macrophages that are already strongly polarized to a classical “M1-like state”. However, we have implemented our results with IL-10- stimulated macrophages, showing that NAA works similarly and in addition to IL-10 in skewing M ϕ macrophages toward an M2-like phenotype. We have selected IL-10, rather than IL4/IL13 because IL-10 is the most representative cytokine in ascitic fluid (Giuntoli *et al*, 2009; Reinartz *et al*, 2014; Takaishi *et al*, 2010). Furthermore the ascitic TAM phenotype can be mimicked by that obtained by IL-10 stimulation (Steitz *et al*, 2020). Finally, IL-10 but not IL-4/IL-13 regulates GS as we reported before (Palmieri *et al*, 2017).

2. The analyses performed for figure 4a and 4b was not clarified in sufficient detail in the manuscript. In my opinion, the correlations on figure 4a and 4b are marginal.

Regarding Figures 4a and 4b, now Figures 5A and B, we understand that correlations, i.e. -0.22 and 0.34, might appear low, yet three factors should be considered. First, here the analyses move from cell models to *in vivo*, implying larger biological variability, spamming, but not limited to, cell composition in each tumor specimen as well as genetic and metabolic variability among different tumor mixtures. Second, the quantification of both tumor purity and M2-like macrophages is based on gene expression data from NGS experiments, which are also affected by technical variability. Third, in spite of those

sources of variation, the two correlations are statistically significant, showing p-values that are not really marginal, i.e. $\sim 5 \times 10^{-4}$ and $\sim 4 \times 10^{-8}$. Finally, the last update of the software we used (Timer2.0) reports, as a “best case”, the expression level of PDCD1, which is known to be expressed in T cells and, in fact, it negatively correlates with tumor purity with a correlation coefficient (-0.27), which is very similar to that observed for GLUL (-0.22). The main text has been modified to provide more details (and reference) about the analyses conducted for the two figures.

3. Figure 4h and I look promising. However, the correlation and p value for figure 4i should be provided. Similarly, Figure 4j and k should also be presented with those information. This is of particular importance since the authors group stage 1 and 2 vs stage 3 and 4. Is there strong correlation along with stage of ovarian cancer?

We have included the correlation and p-values for Figure 4H-I (now in Figure 6C-D) and for Figure 4J-K (now Figure 6E-F). We have plotted data as a function of stage to evaluate the correlation along with stage.

4. Extended from the previous point, it will also important to examine GLUL expression in tumors of EOC patients used in Figure 4. This will strengthen whether the expression of GLUL in tumor negatively associates with IL10 expression and GLUL expression in macrophages.

We agree with the reviewer on the importance of examining GLUL expression in cancer cells. To do so, we have applied two different strategies:

- we have applied a High Resolution CIBERSORTx algorithm to obtain expression profiles from each microenvironment cell as well as from ovarian tumor entity. We have built a signature matrix from GSE146026 single-cell RNA sequencing data which included tumor, macrophages and fibroblast cells. Then we have imputed cell-specific expression profiles from TGCA data including 380 ovarian cancer samples using FPKM normalization, in order to extract GLUL expression levels from each cell type. By doing so, we were able to compare GLUL levels in macrophages and cancer cells, concluding the macrophages are the main contributors to GLUL levels in ovarian cancer.

- we have immunostained ovarian cancer tissues (Figure 5G-I) with specific antibodies for CD163 (as TAM marker) and GS on one side, and Ki67 (as non-selective cancer cell marker) and GLS on the other side. Our results show that the $GLS^{+}Ki67^{+}$ and the $GS^{+}CD163^{+}$ cells increase with stage. Additionally, the number of $CD163^{+}GS^{+}$ cells

significantly correlates with Ki67⁺GLS⁺ in both low and high stages (Figure 5J). These results clearly indicate that GS is mainly expressed in TAMs.

Referee #3:

The data provided using the ovarian cell lines shows that glutamine addiction results in substantial depletion of glutamine in the cultures. The co-culture figures are consistent with the depletion of glutamine by the tumor cells during co-culture and the known effects of Gln deprivation on macrophages. However, the NAA production by the tumor lines only occurs in one of the lines that depletes glutamine, but not the other. Thus the negative correlation between the ability to make Gln and the ability to deplete it from the culture fits well, and the tumor functional assays fit with the availability of Gln but not the production of NAA.

In the revised version of our manuscript, we provide evidence that acquired glutaminolysis by cancer cells, which is associated to ovarian cancer invasiveness, links to NAA production due to a metabolic rewiring. We show that GLUL deficient ovarian cancer cells display high citrate levels, probably due to ammonia accumulation that lowers TCA cycle dehydrogenases flux (Figure 3I). In the mitochondria, oxaloacetate is rerouted to transamination, facilitated by high Glu levels, leading to Asp synthesis, which is promptly acetylated by NAT8L (significantly upregulated in OVCAR3-shGLUL cells compared to OVCAR3 cells, Figure 3G). In the cytosol, citrate is converted to oxaloacetate due to ATP-citrate lyase upregulated in OVCAR3-shGLUL cells compared to OVCAR3 cells (Figure 3H). Glu accumulation due to lower GS activity facilitates oxaloacetate transamination to Asp, which is then acetylated. These results point to the metabolic rewiring linked to high GLS/GS ratio as a main contributor to NAA synthesis. As a further supporting evidence, SKOV3 cells, which are highly glutaminolytic but retain a low but detectable GS protein level (Figure 1A), release NAA at later time points (Figure 3B). In contrast, GS overexpressing SKOV3 cells strongly reduce their ability to produce NAA (Figure S1C).

In vivo Gln is very abundant in blood making depletion unlikely and macrophage "M2" phenotypes, they argue, correlate with Gln availability and GS expression, but these do not correlate with NAA production by the tumor. I understand the argument that high grade tumors have high NAA and that may correlate with increased expression of "M2" phenotypes but the data in Figure 3 are all generated in the absence of Gln. If I am not misunderstanding the data, this seems to be a disconnect. It may be true that NAA drives an M2 like phenotype in the absence of Gln and that Gln deprivation drives an M2

phenotype and that Gln deprivation drives aggressiveness but the data don't fully support a connection between these findings.

We agree that Gln is very abundant in blood. However, it has also been documented that glutamine depletion may occur in the TME (Ishak Gabra *et al*, 2020; Jiang *et al*, 2019; Pan *et al*, 2016). In the revised version of our manuscript we have provided evidence that NAA drives an M2-like phenotype also in the presence of Gln. A comparison of the effects of NAA treatment on M1-like macrophages in presence or absence of Gln clearly indicates that NAA induces a rewiring effect on M1-like macrophages in both conditions, although the increase of CD163, CD206 and GLUL transcript levels is more pronounced when Gln is absent (Figure S2F-I). Furthermore, we have shown that there is a causative link between low GS levels (as in SKOV3 cells) and NAA production, since the acquired glutaminolysis (due to lower GS levels) leads to NAA synthesis (Figure 3B and Figure S1C) while depleting extracellular media of Gln (Figure 1D). In this way, the connection between the described findings is as follows: acquired Gln dependence due to low GS levels rewires cancer cell metabolism leading to NAA production, which is released and drives a M2-like phenotype in macrophages.

If M2 macrophages are "the main contributors to GLUL expression level variations" in tumors then what is the connection to the first figures of this paper?

We have modified this statement. As indicated above, the present version of the manuscript demonstrates that acquired Gln dependence due to low GS levels rewires metabolism leading to NAA production in cancer cells, which is released and drives a M2-like phenotype in macrophages.

Figure 4I and 4K seem to suggest substantial variability in GLUL expression across tumors. This apparent 80 fold spectrum together with the different levels of GLUL expressed in the two lines assessed here do not seem concordant with the conclusion that GLUL variation in the tumors is only due to M2 macrophages. In 4B it would seem there is only about a 4 fold spectrum of "infiltration" so I do not understand how this could lead to an 80 difference in GLUL expression.

We are not sure that comparing GLUL expression levels between cancer tissue, i.e. cell population mixes, and cultured cell lines is appropriate, not only because they are very different biological systems, but also because they are numerically different in their representation. One should expect that the GLUL expression range observed in ~300

samples (tissues) should be higher than that observed in two samples (cell lines). Further, regarding tumor tissues, results suggest that M2-like macrophages contribute to the expression level of GLUL since these two are positively correlating, whereas we cannot exclude that other cell types may contribute to alter GLUL expression. However, it is unlikely that an increase in the fraction of cancer cells may alter the expression level of GLUL, otherwise an increase in tumor purity should result into an increase in GLUL expression, while it is exactly the opposite. The main text has been modified accordingly, to better clarify how these results should be interpreted.

Moreover, although I am not familiar with the analysis in 4B it would seem there is only about a 4 fold spectrum of "infiltration" so I do not understand how this could lead to an 80 difference in GLUL expression.

In the revised version of the manuscript we have clarified the description of Figure 4B, now 5B. The relationship between the two variables, i.e. infiltration level (x) and GLUL expression (y), may be linear with equation $y \sim 8.5x + 7.3$. So, when the infiltration level increases from 0 to 0.2, the GLUL expression level increases from 7.3 to 9, covering most of the expression values of the dataset. We are not considering all the confounding factors (both biological and technical) that move points away from a straight line, further increasing the GLUL expression range.

In my opinion the possibility that NAA alters macrophage phenotype is very interesting but the proposed mechanism, as an antagonist of NMDAR, is not particularly well supported. In figure 3, the NMDA agonist and inhibitor seem to have tremendous effects on their own (especially in the promotion of CD80 expression). Are there endogenous, constitutively expressed ligands in this system? There are no data regarding expression of the receptor and no demonstration of the expected biochemistry of the receptor when NMDA is added. The cited papers provide little comfort. KIM et al (2017) shows a different NMDA antagonist (agmatine) is neuroprotective but the reference also says it can block nos so it is not clear it is working via the receptor. Li et al (2016) shows that THP1 cells treated with PMA apparently can express the NMDAR and that MK801 suppresses expression of the "M1" like cytokine TNF. The latter supports the story here but because this is such an interesting idea, I feel the details of the receptor must be well documented here.

We have provided literature evidence regarding NMDAR in cells of the immune system, from which it can be inferred that NMDAR antagonism induces an M2-like phenotype (Nowak et al, 2019). Furthermore, given the availability of NMDAR reference structures

crystallized in complex with Glu, Gly (ligands present in the system), MK801 and other ligands important for NMDAR activity (Lee *et al*, 2014; Zhu *et al*, 2016), we have produced a computational analysis for estimating NAA binding interactions with NMDAR catalytic binding region, to analyze the relationship between NAA binding and NMDAR activation (Figure S3).

The docking of NAA in the human NMDAR structure shows that NAA binds to the Glu agonist binding region similarly to NMDA and Glu. By comparing residues involved in the binding of Glu and NAA, it is observed that both ligands bind to the same binding pocket residues. However, at variance with NMDA, both Glu and NAA display short-range (below 3.5 Å) direct binding interaction with H485, but only Glu shows short-range binding interactions with Y730. It is known that H485 and Y730 synergically play together a key role in the conformational changes at GluN2 subunit after Glu binding (leong Tou *et al*, 2014), at variance with the shorter aspartate and NMDA, which cannot establish direct interactions with none of them, making both ligands less efficient agonists (Erreger *et al*, 2007) compared to Glu. Direct interactions of NAA with H485 confers to NAA a great affinity for the GluN2 subunit agonist binding domain, compared to NMDA or Asp. However, the shown NAA direct binding interactions uniquely with H485 (known to mediate NMDAR conformational changes necessary for Ca²⁺ influx (leong Tou *et al*, 2014), determine a sort of asymmetry in GluN2 binding interactions, compared to binding interactions observed in presence of Glu, resulting in more rigid conformational changes at the GluN2 binding core in presence of NAA. This means that NAA binds NMDAR with high affinity, but its structure might interfere with Ca²⁺ influx, impairing the signal transduction associated to NMDAR binding. As a support of this hypothesis, we show at a single cell level that NAA does not elicit Ca²⁺ uptake in M1-like macrophages, at variance with NMDA (Figure 4I). These findings, further corroborated by the evidence excluding NAA influx (Figure 4F) and catabolism in macrophages (Figure 4G), support a functional rewiring role of NAA on macrophages through NMDAR antagonism.

The revised version of the manuscript is implemented with this study.

There is a consistent lack of controls in the macrophage experiments. The phenotypes of the beginning populations are not shown making several figures hard to fully interpret. For example Figure 2 and Figure 3C-H are all stimulated, there are no resting controls to compare.

The expectation is that the levels of CD206 etc. are extraordinarily low in LPS and Ifn stimulated cells making 10 fold increases of potentially negligible relevance. The resting levels must be shown in these experiments and in the co-culture experiments as well.

We have included the requested controls in the revised version of the manuscript.

Moreover, the key markers here are easily assessed on the cell surface by FACS yet the data are nearly all qPCR and there are no ELISAs for the chemokines.

A representative evaluation of TNF α with ELISA (Figure 4D) has been included in the revised version of the manuscript.

Lastly, the authors suggest that rather than working as a metabolite that the NAA is a receptor antagonist. Despite this, when they knockdown ASPA there are substantial increases in CD163 mRNA. Although the levels of ASPA are never shown and the efficacy of the knockdown is not shown, the authors conclude "the mechanism of NAA action is not linked to its metabolic processing".

ASPA silencing is known to skew macrophages toward a M2-like phenotype (Gautier *et al*, 2014). Moreover, ASPA is known to have nuclear functions that are independent from NAA catabolism (Hershfield *et al*, 2006). As requested by this Reviewer, we have now included the qPCR evaluation of ASPA mRNA in silenced macrophages (Figure S2A) to assess the efficacy of knockdown. To sustain the hypothesis that NAA is not metabolized by macrophages, we have shown that incubation of macrophages with a concentration of NAA (20 μ M) that is double compared to what used in the experiments of Figure 4 (10 μ M) does not increase the intracellular levels of this molecule (Figure 4G). In the revised version of the manuscript extracellular NAA is measured 24 h after its addition (10 μ M) to M1-like macrophages. As indicated in Figure 4F, NAA levels measured in each condition are never significantly different from 10 μ M. To further support that ASPA is not involved in mediating NAA polarizing effect on macrophages, in the revised version of the manuscript we show that, unlike NAA, ASPA transcript levels in ascitic TAMs do not correlate with tumor stage (Figure S4). Overall, we believe that, in presence of IL-10, the polarizing action of NAA goes through the NMDAR but this does not exclude that, in other contexts, ASPA can affect macrophage functions, both dependently and independently of its

enzymatic activity. This part has been added in the discussion and the sentence reported by the reviewer has been toned down and placed in the context of our study.

MINOR CONCERNS

Similarly, how are the ascites-derived macrophages isolated and handled. The data refer to values relative to M0. What does that mean?

In the revised version of the manuscript, more details are included regarding isolation of macrophages and selection of the corresponding controls.

It is assumed that the NAA is very acidic. Acidic conditions have been shown to favor alternative activation of macrophages. Has this been ruled out?

We have checked the pH changes following NAA addition to RPMI medium. At the concentrations used in the experiments, we did not record significant variations in pH.

What is the glutamine level in the ascites?

We have measured glutamine in the ascites and include this result in the revised version of the manuscript (Figure S4). Gln levels do not change at the increasing of the stage. This is expected as Gln concentration in the ascites represents the net result of secretion (by TAMs) and uptake (by cancer cells).

As noted, NAA levels and expression of Nat8l has already been associated with poor outcomes by Zand et al. thus the statement "Our work paves the way towards the possibility to explore NAA as a novel, predictive biomarker for disease progression in ovarian cancer." Is not really appropriate.

We have changed this sentence in the revised version of the manuscript.

Figure 1F should show the effects of supplementation of Gln. Given the growth differences it could be lactate etc.

We agree that in conditions of Gln depletion, cells rely more on glycolysis leading to increased lactate production. Exogenous lactate levels increase Gln dependence (Pérez-Escuredo *et al*, 2016), thus stressing cells even more. We will include this comment in the Discussion.

Different gene sets are used for different figures for M1 and M2. CD206 should be shown in figure 2

This gene is included in the revised version of Figure 2.

Figure 3 also should show re-introduction of glutamine

In order to clearly comprehend the role of Gln in the rewiring of M1-like macrophages by NAA, we have compared the changes in M1 and M2-like markers following NAA and NAA/IL-10 and NAA/IL-10/NMDA treatments on macrophages cultured in Gln rich or depleted medium. As indicated in Figure S2, the changes in the levels of M2-markers is higher when Gln is absent, whereas CD80 is higher in Gln-rich medium. This further corroborates the notion that Gln depletion works synergically with NAA in rewiring macrophages toward a M2-like phenotype.

What is the status of ASPA and Nat8l in the macs and tumors?

As note in Figure S4, included in the revised version of the manuscript, ASPA transcript levels do not change in ascitic macrophages at the increase of the stage, indicating that NAA catabolism is not relevant for the acquisition of a pro-tumoral macrophage phenotype. NAT8L transcript levels were too low to be evaluated (data not show). Regarding cancer cells, NAT8L positively correlates with Ki67 (Figure 5), indicating that NAA synthesis directly or indirectly mark an active proliferative state.

Figure 3C shows only a slight increase of CD206 with NAA+IL10 compared to these agents alone. This is not even additive, much less synergistic as suggested. The effect on CD163 is perhaps additive.

We have changed this statement in the revised version of the manuscript.

In figure 3 it is not clear why the IL10 and MK108 combo was tested (with or without NMDA agonist)? This is superfluous data that doesn't add to the experiment and makes a confusing chart even more confusing.

We have removed that condition from the revised Figure 4.

There are multiple references to "M2 skewing" and "polarizing to toward 'M2-like'" and "M1 to M2 switch" but as noted above the controls are not shown.

Is this really just interfering with establishment of M1?

We have included the controls to provide a more precise description of the effect of NAA on macrophages. Since NAA is added 24 hours after LPS/IFN γ addition, NAA can be considered a molecule rewiring the phenotype of established M1-like macrophages.

Figure 4A is not convincing

We understand that correlations, i.e. -0.22 and 0.34, might appear low, yet three factors should be considered. First, here the analyses move from cell models to *in vivo*, that is to systems that undergo biological variability, including, but not limited to, the cell type variability in each tumor mixture and the genetic and metabolic variability among different tumor mixtures. Second, the quantification of both tumor purity and M2 macrophage is based on gene expression data from NGS experiments, which are also affected by technical variability. Third, in spite of those sources of variation, the two correlations are statistically significant, showing p-values that are not really marginal, i.e. $\sim 5 \times 10^{-4}$ and $\sim 4 \times 10^{-8}$. Finally, the last update of the software we used (Timer2.0) reports, as a “best case”, the expression level of PDCD1, which is known to be expressed in T cells and, in fact, it negatively correlates with tumor purity with a correlation coefficient (-0.27), which is very similar to that observed for GLUL (-0.22). The main text has been modified to provide more details (and reference) about the analyses conducted for the two figures.

Contrary to the contentions of these authors there are multiple animal models of ovarian cancer. In particular, because the proposed model involves modification of macrophage phenotypes, information could be derived from xenografts of these human lines into immunocompromised mice. These sorts of studies are not necessarily needed here but paucity of models should not be suggested.

We have corrected our statement in the revised version of the manuscript.

References:

- Erreger K, Geballe MT, Kristensen A, Chen PE, Hansen KB, Lee CJ, Yuan H, Le P, Lyuboslavsky PN, Micale N *et al* (2007) Subunit-specific agonist activity at NR2A-, NR2B-, NR2C-, and NR2D-containing N-methyl-D-aspartate glutamate receptors. *Mol Pharmacol* **72**: 907–920
- Gautier EL, Ivanov S, Williams JW, Huang SCC, Marcelin G, Fairfax K, Wang PL, Francis JS, Leone P, Wilson DB *et al* (2014) Gata6 regulates aspartoacylase expression in

resident peritoneal macrophages and controls their survival. *J Exp Med* **211**: 1525-1531.

Giuntoli RL, Webb TJ, Zoso A, Rogers O, Diaz-Montes TP, Bristow RE, Oelke M (2009) Ovarian cancer-associated ascites demonstrates altered immune environment: Implications for antitumor immunity. *Anticancer Res* **29**: 2875-2884.

Hershfield JR, Madhavarao CN, Moffett JR, Benjamins JA, Garbern JY, Namboodiri A (2006) Aspartoacylase is a regulated nuclear-cytoplasmic enzyme. *FASEB J* **20**: 2139-2141

leong Tou W, Chang S Sen, Wu D, Lai TW, Wang YT, Hsu CY, Yu-Chian Chen C (2014) Molecular level activation insights from a NR2A/NR2B agonist. *J Biomol Struct Dyn* **32**:683-693

Ishak Gabra MB, Yang Y, Li H, Senapati P, Hanse EA, Lowman XH, Tran TQ, Zhang L, Doan LT, Xu X *et al* (2020). Dietary glutamine supplementation suppresses epigenetically-activated oncogenic pathways to inhibit melanoma tumour growth. *Nat Commun* **11**: 3326

Jiang J, Srivastava S, Zhang J (2019) Starve cancer cells of glutamine: Break the spell or make a hungry monster? *Cancers* **11**: 804

Lee CH, Lü W, Michel JC, Goehring A, Du J, Song X, Gouaux E (2014) NMDA receptor structures reveal subunit arrangement and pore architecture. *Nature* **511**: 191-197

Nowak W, Grendas LN, Sanmarco LM, Estecho IG, Arena ÁR, Eberhardt N, Rodante DE, Aoki MP, Daray FM, Carrera Silva EA, Errasti AE (2019) Pro-inflammatory monocyte profile in patients with major depressive disorder and suicide behaviour and how ketamine induces anti-inflammatory M2 macrophages by NMDAR and mTOR. *EBioMedicine* **50**: 290–305

Palmieri EM, Menga A, Martín-Pérez R, Quinto A, Riera-Domingo C, De Tullio G, Hooper DC, Lamers WH, Ghesquière B, McVicar DW *et al* (2017) Pharmacologic or Genetic Targeting of Glutamine Synthetase Skews Macrophages toward an M1-like Phenotype and Inhibits Tumor Metastasis. *Cell Rep* **20**: 1654-1666

Pan M, Reid MA, Lowman XH, Kulkarni RP, Tran TQ, Liu X, Yang Y, Hernandez-Davies JE, Rosales KK, Li H *et al* (2016). Regional glutamine deficiency in tumours promotes dedifferentiation through inhibition of histone demethylation. *Nat Cell Biol* **18**: 1090-1101

Pérez-Escuredo J, Dadhich RK, Dhup S, Cacace A, Van Hée VF, De Saedeleer CJ, Sboarina M, Rodriguez F, Fontenille MJ, Brisson L *et al* (2016) Lactate promotes glutamine uptake and metabolism in oxidative cancer cells. *Cell Cycle* **15**: 72-83

Reinartz S, Schumann T, Finkernagel F, Wortmann A, Jansen JM, Meissner W, Krause M, Schwörer AM, Wagner U, Müller-Brüsselbach S, Müller R (2014) Mixed-polarization phenotype of ascites-associated macrophages in human ovarian carcinoma: Correlation of CD163 expression, cytokine levels and early relapse. *Int J Cancer* **134**: 32–42

Steitz AM, Steffes A, Finkernagel F, Unger A, Sommerfeld L, Jansen JM, Wagner U, Graumann J, Müller R, Reinartz S (2020) Tumor-associated macrophages promote ovarian cancer cell migration by secreting transforming growth factor beta induced (TGFB1) and tenascin C. *Cell Death Dis* **11**: 249

Takaishi K, Komohara Y, Tashiro H, Ohtake H, Nakagawa T, Katabuchi H, Takeya M

(2010) Involvement of M2-polarized macrophages in the ascites from advanced epithelial ovarian carcinoma in tumor progression via Stat3 activation. *Cancer Sci* **101**: 2128–2136

Zhu S, Stein RA, Yoshioka C, Lee CH, Goehring A, McHaourab HS, Gouaux E (2016) Mechanism of NMDA Receptor Inhibition and Activation. *Cell* **165**: 704-714

Dear Dr. Castegna,

Thank you for submitting your revised manuscript. It has now been seen by two referees. Former referee #3 was unavailable to re-review the manuscript. Therefore, referee #4 was recruited to cover the expertise of referee #3 and to specifically evaluate the points raised by referee #3.

As you can see, the referees find that the study is significantly improved during revision and recommend publication. However, I need you to address the remaining points below before I can accept the manuscript:

- Please address the outstanding concerns of referee #4 by either experimentally or carefully revising the conclusions in the text and provide a point-by-point response, as per suggestions of referee #4.
- As per our guidelines, please add a 'Data Availability Section', where you state that no data were deposited in a public database.
- Please add 'Conflict of Interests' and 'Author Contributions' sections.
- Please complete the funding information in the manuscript submission system.
- We note that the following is currently not called out in the text: panels of Fig S3 and Figs S4C+D.
- Please rename the Table S1 as Dataset EV 1 and update the callouts to it in the text.
- Please rename the supplementary figures as EV Figure 1, 2, 3 etc. Please update their callouts in the text, too.
- The EV figure legends should come after the main figure legends in the Manuscript file.
- Please move the supplementary Methods in to the main Materials & Methods.
- As per our format requirements, the title length cannot exceed 100 characters (including spaces). The title is currently too long.
- We notice that there are two Reagent Tables, however, they do not list reagents. Please rename them as EV Table 1 and 2 and update the callouts in the text. Please include their legends in the tables.
- Papers published in EMBO Reports include a 'synopsis' and 'bullet points' to further enhance discoverability. Both are displayed on the html version of the paper and are freely accessible to all readers. The synopsis includes a short standfirst summarizing the study in 1 or 2 sentences that summarize the paper and are provided by the authors and streamlined by the handling editor. I would therefore ask you to include your synopsis blurb and 3-5 bullet points listing the key experimental findings.
- In addition, please provide an image for the synopsis. This image should provide a rapid overview of the question addressed in the study but still needs to be kept fairly modest since the image size cannot exceed 550x400 pixels.

Thank you again for giving us to consider your manuscript for EMBO Reports, I look forward to your minor revision.

Kind regards,

Deniz Senyilmaz Tiebe

--

Deniz Senyilmaz Tiebe, PhD

Referee #2:

The revised manuscript address most of my concerns. Based on the content and the newly included results, this manuscript provides interesting findings based on solid data. Therefore, I would recommend for publication.

Referee #4:

In this study by Menga et al, the authors report that ovarian cancer cells with low glutamine synthetase (GS) expression induce GS expression macrophages via secretion of n-acetylaspartate, resulting M2-like polarization. This is an interesting finding that complements the previous work by the authors showing that silencing GS in macrophages induces an M1-like phenotype.

In the revised version of the manuscript, Menga et al have addressed most of the original referee #3's concerns. However, some concerns remain, mainly regarding data interpretation and validation, that need to be addressed to support publication.

- 1) The model in Figure 3I by which the authors explain the connection of glutamine addiction and NAA production is flawed by the fact that only mRNA expression levels of metabolic enzymes and one-time point measurements of a few individual metabolites was used to construct the model. The data presented support but do not prove their model. mRNA expression levels are not a surrogate for enzymatic activity. The authors should be more careful in discussing their potential model in light of the data they have. Alternatively, they need to perform appropriate flux experiments to confirm it, i.e. that low GS/increased glutaminolysis drives glycolytic carbon into NAA synthesis.
- 2) The correlative analyses in Figures 5 and 6 are interesting but should not be used by the authors as "validation", "confirmation" or "proof" of the experimental data and the proposed mechanism. These data support the authors' claims, but do not provide causation. The authors will need to rephrase their conclusions.
- 3) The absence of a change in extra- and intracellular NAA levels after NAA supplementation to macrophages suggests, but does not exclude, the absence of NAA influx and catabolism in macrophages. The authors should either provide evidence by tracing labeled NAA into potential degradation products in macrophages, or word their statement more carefully.
- 4) In agreement with original referee #3, the authors should perform flow cytometry analysis of macrophage polarization markers using established cell surface markers. This should be easy to do and is needed to support the authors' main conclusion.

Response to reviewer 4

Here is the response to Reviewer # 4's criticisms.

In this study by Menga et al, the authors report that ovarian cancer cells with low glutamine synthetase (GS) expression induce GS expression macrophages via secretion of n-acetylaspartate, resulting M2-like polarization. This is an interesting finding that complements the previous work by the authors showing that silencing GS in macrophages induces an M1-like phenotype.

In the revised version of the manuscript, Menga et al have addressed most of the original referee #3's concerns. However, some concerns remain, mainly regarding data interpretation and validation, that need to be addressed to support publication.

1) The model in Figure 3I by which the authors explain the connection of glutamine addiction and NAA production is flawed by the fact that only mRNA expression levels of metabolic enzymes and one-time point measurements of a few individual metabolites was used to construct the model. The data presented support but do not prove their model. mRNA expression levels are not a surrogate for enzymatic activity. The authors should be more careful in discussing their potential model in light of the data they have. Alternatively, they need to perform appropriate flux experiments to confirm it, i.e. that low GS/increased glutaminolysis drives glycolytic carbon into NAA synthesis.

The link between glutaminolysis and NAA synthesis is extensively described in cancer studies by using tracing experiments, providing strong evidence that glutaminolysis drives aspartate synthesis for NAA production (Lou *et al*, 2016; Wynn *et al*, 2016). Additionally, conditions increasing GS expression are reported to reduce NAA synthesis (Wynn *et al*, 2016). We have included and discussed these references in the Discussion and also we have downtoned our interpretation of the metabolic data regarding NAA synthesis.

2) The correlative analyses in Figures 5 and 6 are interesting but should not be used by the authors as "validation", "confirmation" or "proof" of the experimental data and the proposed mechanism. These data support the authors' claims, but do not provide causation. The authors will need to rephrase their conclusions.

We have changed the conclusions according to the reviewer's suggestions.

3) The absence of a change in extra- and intracellular NAA levels after NAA supplementation to macrophages suggests, but does not exclude, the absence of NAA influx and catabolism in macrophages. The authors should either provide evidence by tracing labeled NAA into potential degradation products in macrophages, or word their statement more carefully.

In the original and the revised version of the manuscript we have provided evidence that the extracellular levels of NAA does not change during the experimental time frame. In the present version of the manuscript, we are providing a time course intracellular quantification of NAA levels following a 24 and 48 h incubation with increasing NAA levels (10 and 20 μ M). This in our opinion strengthens even further that NAA does not cross the cellular membrane. Since we cannot rule out that at higher concentrations and/or at longer time points this might occur, we have modified the Discussion to consider this possibility.

We have also enriched the Discussion section providing literature evidence that ASPA expression does not correlate with tumor NAA levels in ovarian cancer samples (Zand *et al*, 2016) and that a non-catabolic role of NAA has been previously hypothesized (Bogner-Strauss, 2017).

4) In agreement with original referee #3, the authors should perform flow cytometry analysis of macrophage polarization markers using established cell surface markers. This should be easy to do and is needed to support the authors' main conclusion.

In the present version of the manuscript we have included the flow cytometry analysis of macrophage polarization markers following NAA treatment as suggested by the reviewer (current EV Figure 2).

References

- Bogner-Strauss JG (2017) N-Acetylaspartate Metabolism Outside the Brain: Lipogenesis, Histone Acetylation, and Cancer. *Front Endocrinol (Lausanne)* 8: 240
- Lou T-F, Sethuraman D, Dospoy P, Srivastva P, Kim HS, Kim J, Ma X, Chen P-H, Huffman KE, Frink RE, *et al* (2016) Cancer-Specific Production of N-Acetylaspartate via NAT8L Overexpression in Non-Small Cell Lung Cancer and Its Potential as a Circulating Biomarker. *Cancer Prev Res (Phila)* 9: 43–52
- Wynn ML, Yates JA, Evans CR, Van Wassenhove LD, Wu ZF, Bridges S, Bao L, Fournier C, Ashrafzadeh S, Merrins MJ, *et al* (2016) RhoC GTPase Is a Potent Regulator of Glutamine Metabolism and N-Acetylaspartate Production in Inflammatory Breast Cancer Cells. *J Biol Chem* 291: 13715–13729
- Zand B, Previs RA, Zacharias NM, Rupaimoole R, Mitamura T, Nagaraja AS, Guindani M, Dalton HJ, Yang L, Baddour J, *et al* (2016) Role of Increased n-acetylaspartate Levels in Cancer. *Journal of the National Cancer Institute* 108: djv426–djv426

Dear Dr. Castegna,

Thank you for submitting your revised manuscript. I have now looked at everything and all is fine. Therefore, I am very pleased to accept your manuscript for publication in EMBO Reports.

Congratulations on a nice work!

Kind regards,

Deniz Senyilmaz Tiebe

--

Deniz Senyilmaz Tiebe, PhD
Editor
EMBO Reports

--

At the end of this email I include important information about how to proceed. Please ensure that you take the time to read the information and complete and return the necessary forms to allow us to publish your manuscript as quickly as possible.

As part of the EMBO publication's Transparent Editorial Process, EMBO reports publishes online a Review Process File to accompany accepted manuscripts. As you are aware, this File will be published in conjunction with your paper and will include the referee reports, your point-by-point response and all pertinent correspondence relating to the manuscript.

If you do NOT want this File to be published, please inform the editorial office within 2 days, if you have not done so already, otherwise the File will be published by default [contact: emboreports@embo.org]. If you do opt out, the Review Process File link will point to the following statement: "No Review Process File is available with this article, as the authors have chosen not to make the review process public in this case."

Should you be planning a Press Release on your article, please get in contact with emboreports@wiley.com as early as possible, in order to coordinate publication and release dates.

Thank you again for your contribution to EMBO reports and congratulations on a successful publication. Please consider us again in the future for your most exciting work.

THINGS TO DO NOW:

You will receive proofs by e-mail approximately 2-3 weeks after all relevant files have been sent to our Production Office; you should return your corrections within 2 days of receiving the proofs.

Please inform us if there is likely to be any difficulty in reaching you at the above address at that time. Failure to meet our deadlines may result in a delay of publication, or publication without your corrections.

All further communications concerning your paper should quote reference number EMBOR-2020-51981V3 and be addressed to emboreports@wiley.com.

Should you be planning a Press Release on your article, please get in contact with emboreports@wiley.com as early as possible, in order to coordinate publication and release dates.

Corresponding Author Name: Alessandra Castegna and Massimiliano Mazzone

Journal Submitted to: EMBO REPORTS

Manuscript Number: